# Hydrophobic recognition allows the glycosyltransferase UGT76G1 to catalyze its substrate in two orientations

Ting Yang[1,8], Jinzhu Zhang[1,8], Dan Ke[1,8], Wenxian Yang[1,8], Minghai Tang[2], Jian Jiang[1], Guo Cheng[3], Jianshu Li [4], Wei Cheng [2], Yuquan Wei[2], Qintong Li [5], James H. Naismith [2,6,7] & Xiaofeng Zhu [1]

Diets high in sugar are recognized as a serious health problem, and there is a drive to reduce their consumption. Steviol glycosides are natural zero-calorie sweeteners, but the most desirable ones are biosynthesized with low yields. UGT76G1 catalyzes the β (1–3) addition of glucose to steviol glycosides, which gives them the preferred taste. UGT76G1 is able to transfer glucose to multiple steviol substrates yet remains highly specific in the glycosidic linkage it creates. Here, we report multiple complex structures of the enzyme combined with biochemical data, which reveal that the enzyme utilizes hydrophobic interactions for sub-strate recognition. The lack of a strict three-dimensional recognition arrangement, typical of hydrogen bonds, permits two different orientations for β (1–3) sugar addition. The use of hydrophobic recognition is unusual in a regio- and stereo-specific catalysis. Harnessing such non-specific hydrophobic interactions could have wide applications in the synthesis of complex glycoconjugates.

[1] Key Laboratory of Bio-Resource and Eco-Environment of Ministry of Education, College of Life Sciences, Sichuan University; State Key Laboratory of Biotherapy and Cancer Center, West China Hospital, Sichuan University, 610064 Chengdu, China. [2] State Key Laboratory of Biotherapy and Cancer Center, West China Hospital, Sichuan University, 610041 Chengdu, China. [3] West China School of Public Health, Healthy Food Evaluation Research Center and State Key Laboratory of Biotherapy and Cancer Center, Sichuan University, 610041 Chengdu, China. [4] Department of Biomedical Polymers and Artificial Organs, College of Polymer Science and Engineering and State Key Laboratory of Polymer Materials Engineering, Sichuan University, 610065 Chengdu, China. [5] Department of Pediatrics, Obstetrics and Gynecology, West China Second University Hospital, Key Laboratory of Birth Defects and Related Diseases of Women and Children, Ministry of Education, Sichuan University, 610041 Chengdu, China. [6] Division of Structural Biology, Wellcome Trust Centre of Human Genomics, Oxford OX3 7BN, UK. [7] Rosalind Franklin Institute, Harwell Campus, Didcot OX11 0FA, UK. [8] These authors contributed equally: Ting Yang, Jinzhu Zhang, Dan Ke, Wenxian Yang. Correspondence and requests for materials should be addressed to Q.L. (email: liqintong@scu.edu.cn) or to J.H.N. (email: naismith@strubi.ox.ac.uk) or to X.Z. (email: zhuxiaofeng@scu.edu.cn)

The excessive dietary consumption of sugars such as glucose, fructose, and sucrose is recognized to have adverse health consequences that include obesity, diabetes, and cancer[1,2]. Many countries face a serious public health burden from such high-calorie sweeteners[3–5]. The World Health Organization has supported actions taken by some countries, such as sugar taxes, to reduce sugar consumption[1]. It is the sweet taste, imparted by the binding of sugar to the G-protein coupled receptors hTAS1R2/hTAS1R3 in the taste buds[6,7], that is highly prized by humans. Sweet food with quick-release energy is highly beneficial in a calorie-restricted diet and therefore selected by evolution[8]. The reduction of sugar consumption by substitution with low- or zero-calorie sweeteners is a favored strategy. Saccharine and aspartame are well-established artificial sweeteners but have not displaced sugar. The reasons are actively debated but include organoleptic properties, the human perception of the sweet taste, and the perception of health risks[9–11]. There is a demand for a 'natural' low-calorie sugar substitute that has a desirable sweetness.

The leaves of *Stevia rebaudiana* Bertoni, a South American shrub, have been used as a natural sweetener by the indigenous population for centuries[12,13]. The sweetness originates from a mixture of steviol glycosides found in the leaves. Steviol glycosides consist of a common diterpenoid steviol backbone (aglycone) and a variable glycone composed mainly of glucose molecules (the steviol glucosides). Glucose molecules are linked by β-glycosidic bonds to the steviol aglycone at the C13-hydroxyl (denoted the R1 site) and/or C19-carboxylate (denoted the R2 site) (Fig. 1a). The β-configured glycosidic bonds in these steviol glucosides remain largely undigested by humans, and thus little energy is liberated. The sustained consumption of steviol glucosides has been reported to potentiate TRPM5 ion channel activity and glucose-induced insulin secretion, preventing the development of diabetes in mice[14]. The combination of these properties has made steviol glucosides the attractive sugar substitutes.

The most abundant steviol glucosides in *S. rebaudiana* leaves, stevioside (ST) and rebaudioside A (Reb A) (Fig. 1a) are 200-fold sweeter than sucrose and were the first commercially available steviol sweeteners[13,15]. However, both ST and Reb A also activate the bitter taste receptors hTAS2R4 and hTAS2R14, limiting their acceptance[16]. Two trace steviol glucosides, rebaudioside D (Reb D) and rebaudioside M (Reb M) (Fig. 1a), show higher sweetness potency and a much reduced bitter aftertaste[17,18]. Reb M, in particular, provides a fast, clean sweet taste and is the steviol glucoside most similar to sucrose in organoleptic properties.

The low natural abundance of Reb M and Reb D prevents their mass production by extraction. Genetically modified *S. rebaudiana* with increased production or microbial production of Reb M and Reb D present solutions, but both require a full understanding of the biosynthetic pathway, particularly glycosylation. The biosynthetic pathway of steviol glucosides has been determined, and a series of glycosyltransferases (UGTs) were identified that utilize a nucleoside diphosphate-activated sugar (UDP-glucose, UDPG) as the sugar donor[19] (Fig. 1b). UGT85C2 and UGT74G1 place the first glucose with a β-1 glycosidic linkage at the C13-hydroxyl (R1) and C19-carboxylate (R2) positions of the steviol aglycone respectively. For clarity, we name these two initial glucoses linked directly at the C13-hydroxyl (R1) or C19-carboxylate (R2) as glucose A, followed by R1 or R2 to indicate the position at the steviol aglycone. These two glucoses, AR1 and AR2, can be further modified by UGT91D2 with β (1–2) glucose addition[18] and UGT76G1 with β (1–3) glucose addition[19] (Fig. 1b). The incoming glucose added by UGT91D2 or UGT76G1 to glucose A is named as glucose B or C, similarly followed by R1 or R2 to show the position at the steviol aglycone.

UGT76G1 alone performs β (1–3) glycosylation of Reb D and Reb M in *S. rebaudiana*. We show that it utilizes UDPG to modify the glucoses AR1 and AR2 in a range of steviol substrates, but does not modify free sugars without the diterpenoid steviol ring. The enzyme creates a β (1–3) linkage by recognizing the 3-hydroxyl of the glucoses AR1 and AR2 but neglecting their different structural contexts, in which the glucoses AR1 and AR2 are at the asymmetric ends of the diterpenoid steviol ring. By combining multiple structures of UGT76G1 with the reactivity profile of the available authentic and two self-made steviol glucosides, we have revealed how the enzyme relies on the hydrophobic diterpenoid steviol aglycone to allow flexibility in binding orientation but specificity in catalysis, which provide a rational route for the development of the enzyme in sugar biotechnology.

## Results

**UGT76G1 reacts at two asymmetric positions of the sugar acceptor.** UGT76G1 is a UDP-glucose (UDPG)-dependent glycosyltransferase, where UDPG is the sugar donor and UDP is released after sugar transfer. UGT76G1 was characterized with a range of steviol substrates, and the products were confirmed by comparison with authentic standards using high-performance liquid chromatography (HPLC) (Supplementary Fig. 1) and high-resolution mass spectrometry (MS). Unlike the metal-dependent GT-A fold glycosyltransferases[20], the activity of UGT76G1 was not significantly altered by the addition of $Mg^{2+}$ or the removal of $Mg^{2+}$ by EDTA (Supplementary Fig. 2). UGT76G1 shows no detectable activity (by high-resolution MS) on glucose, the disaccharide AB (sophorose) (Fig. 1a) or the steviol aglycone backbone. UGT76G1 reacts with steviolbioside (STB), Reb A, ST, Reb D, rebaudioside E (Reb E) and rubusoside (Rubu) (Fig. 1a).

Reaction with STB, which has a disaccharide AB at R1, results in a single glycosylation product, rebaudioside B (Reb B) (Fig. 2a and Supplementary Fig. 3). Reb A, which contains a trisaccharide ABC at R1 and a glucose A at the R2 position is converted to rebaudioside I (Reb I) (Fig. 2b and Supplementary Fig. 4). The velocity of substrate consumption and product production show that the rate of the glycosylation to R2 is slower than that to the R1 position. UGT76G1 transfers a glucose to ST to give Reb A, which undergoes a further glycosylation at the R2 position to produce Reb I by a much slower reaction (Fig. 2c and Supplementary Fig. 5). There is no evidence for transfer to R2 without first transfer to R1. Reb E, which has an identical disaccharide AB at both the R1 and R2 positions, is converted to Reb D by transfer at the R1 position. Reb D is further modified by a second glucose transfer at the R2 position to yield Reb M (Fig. 2d and Supplementary Fig. 6). Similar to ST, there is no evidence of any transfer to R2 before the R1 transfer. Rubu has a single glucose A at both the R1 and R2 positions, and incubation with the enzyme gives two products, denoted by RX and RY. The product RX appears quickly with an $[RX–H]^-$ mass of 803.36 Da, indicating a single glucose transfer. The second product RY appears to arise from a subsequent reaction and has an $[RY–H]^-$ mass of 965.42 Da, indicating a second transfer of glucose (Fig. 3). Based on the other steviol substrates, we predicted that the first product RX represented a glucose transfer at R1, and the second product RY resulted from a second glucose transfer at R2. We lacked authentic standards to unambiguously confirm this. Indirect evidence comes from tandem mass spectrometry fragmentation (MS/MS). We reasoned that the more labile ester bond at the R2 site of the aglycone would fragment first. This was established to be correct in all of the authentic steviol standards we had (Supplementary Figs. 3–6). The two products from Rubu were individually purified by HPLC and fragmented by MS/MS. The parent ion of RX gives the most abundant product ion with a

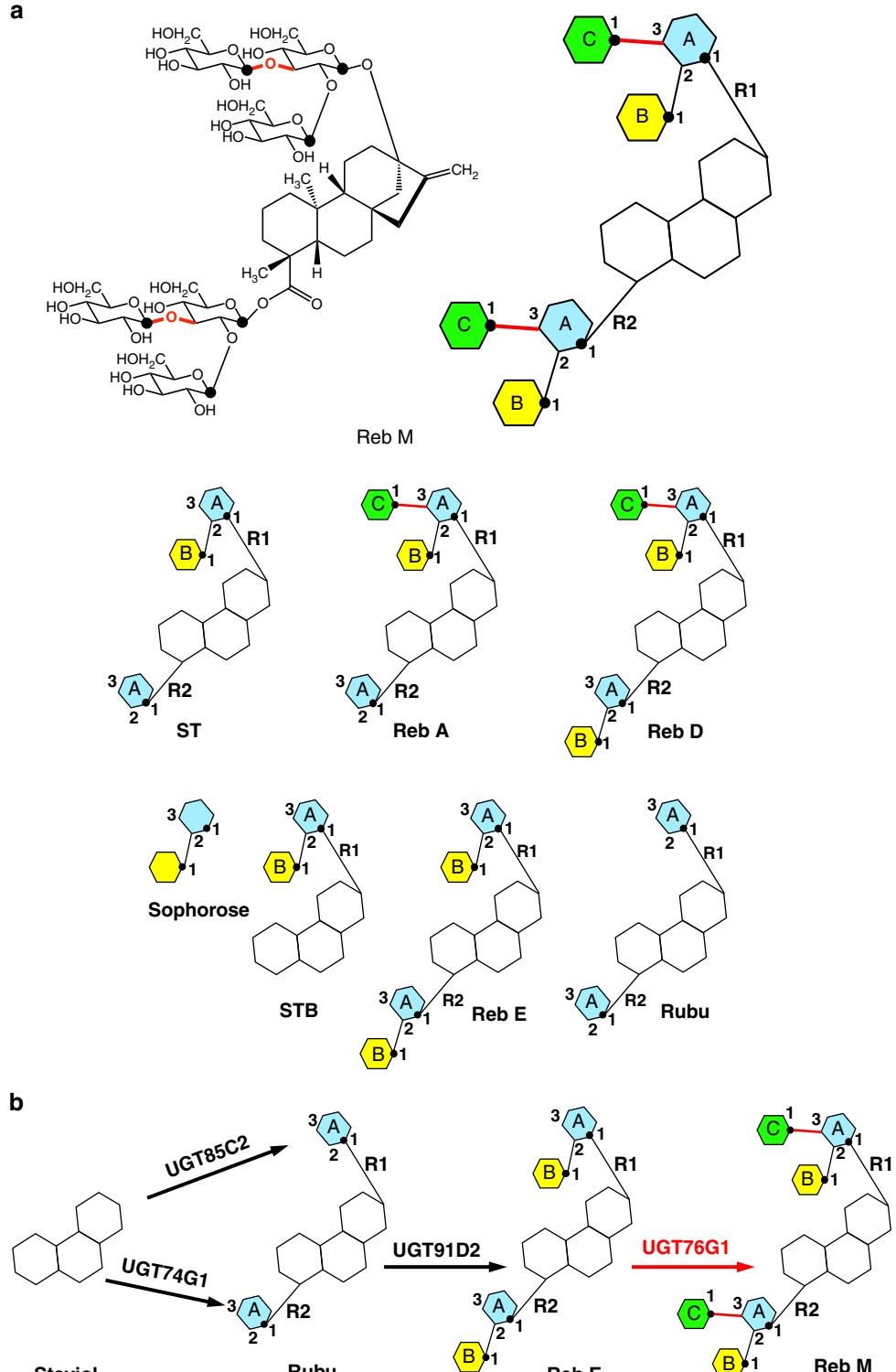

**Fig. 1** The steviol glucosides related to UGT76G1 and the reactions in the biosynthesis pathway of steviol glucoside. **a** The chemical structure of Reb M and cartoon representations of the typical steviol glucosides Reb M, ST, Reb A, Reb D, sophorose, STB, Reb E, and Rubu. In the cartoon representation, the glycone units are represented by individually colored hexagons with the letters A, B, and C for clarity and the position of the 1-hydroxyl of glycone is marked by a black dot. All β (1–3) glycolic bonds formed by UGT76G1 are marked in red. **b** The reactions in the steviol glucoside biosynthesis pathway of *Stevia rebaudiana* Bertoni. The reaction and the glycosidic bonds involving UGT76G1 are labeled in red

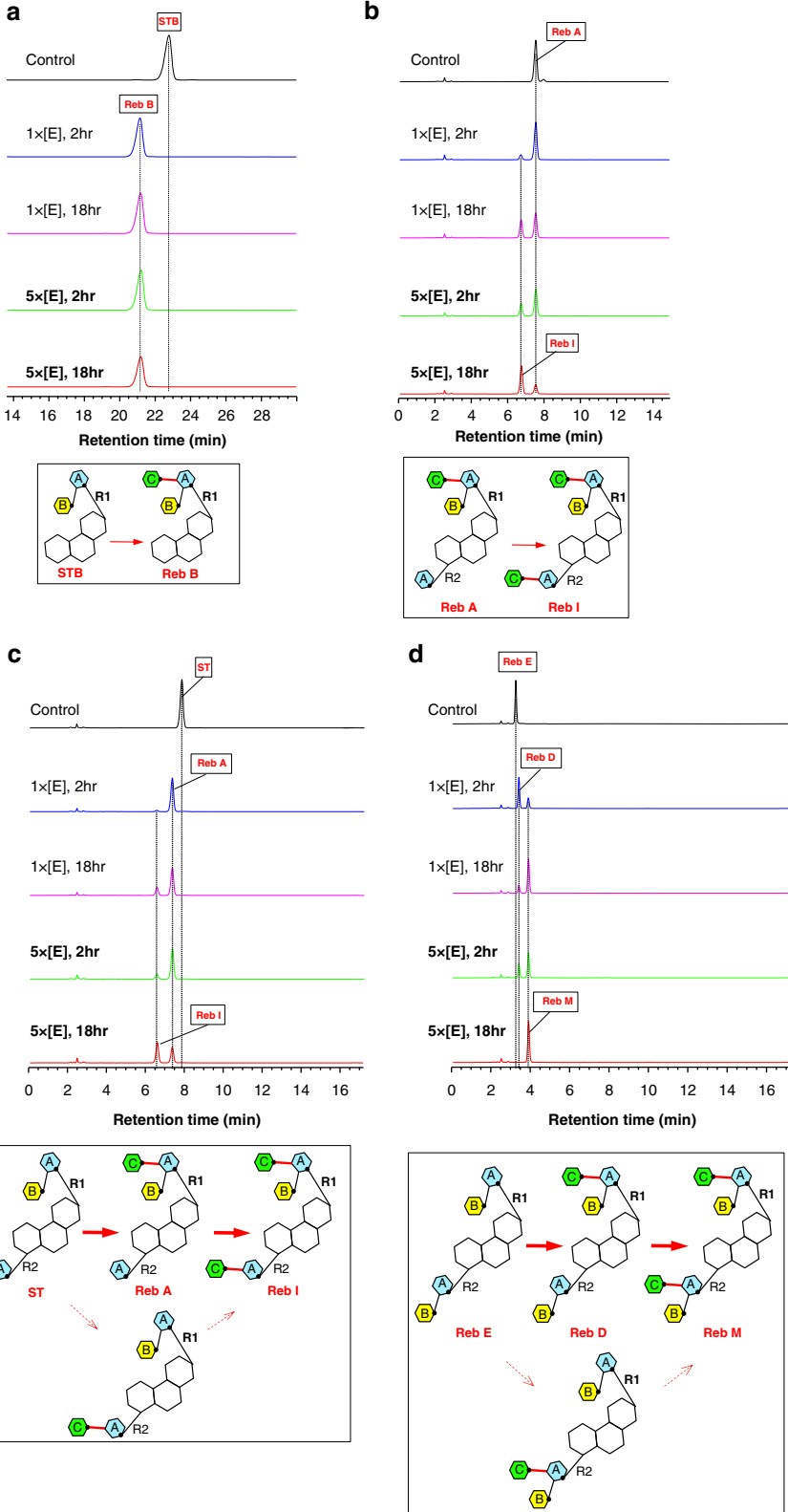

**Fig. 2** The steviol products of the sugar transfers by UGT76G1. **a–d** HPLC traces of the reactions of STB, Reb A, ST, Reb E were analyzed, respectively, by HPLC. The five HPLC traces from top to bottom represent the following reaction conditions: no enzyme for 18 h (black), 0.03 mg ml$^{-1}$ enzyme (1×) for 2 h (blue), 0.03 mg ml$^{-1}$ enzyme (1×) for 18 h (purple), 0.15 mg ml$^{-1}$ enzyme (5×) for 2 h (green) and 0.15 mg ml$^{-1}$ enzyme (5×) for 18 h (red). The product is identified by the authentic standard. The yields of the products are related to the enzyme concentration and the reaction duration. The reactions catalyzed by UGT76G1 are shown in the box

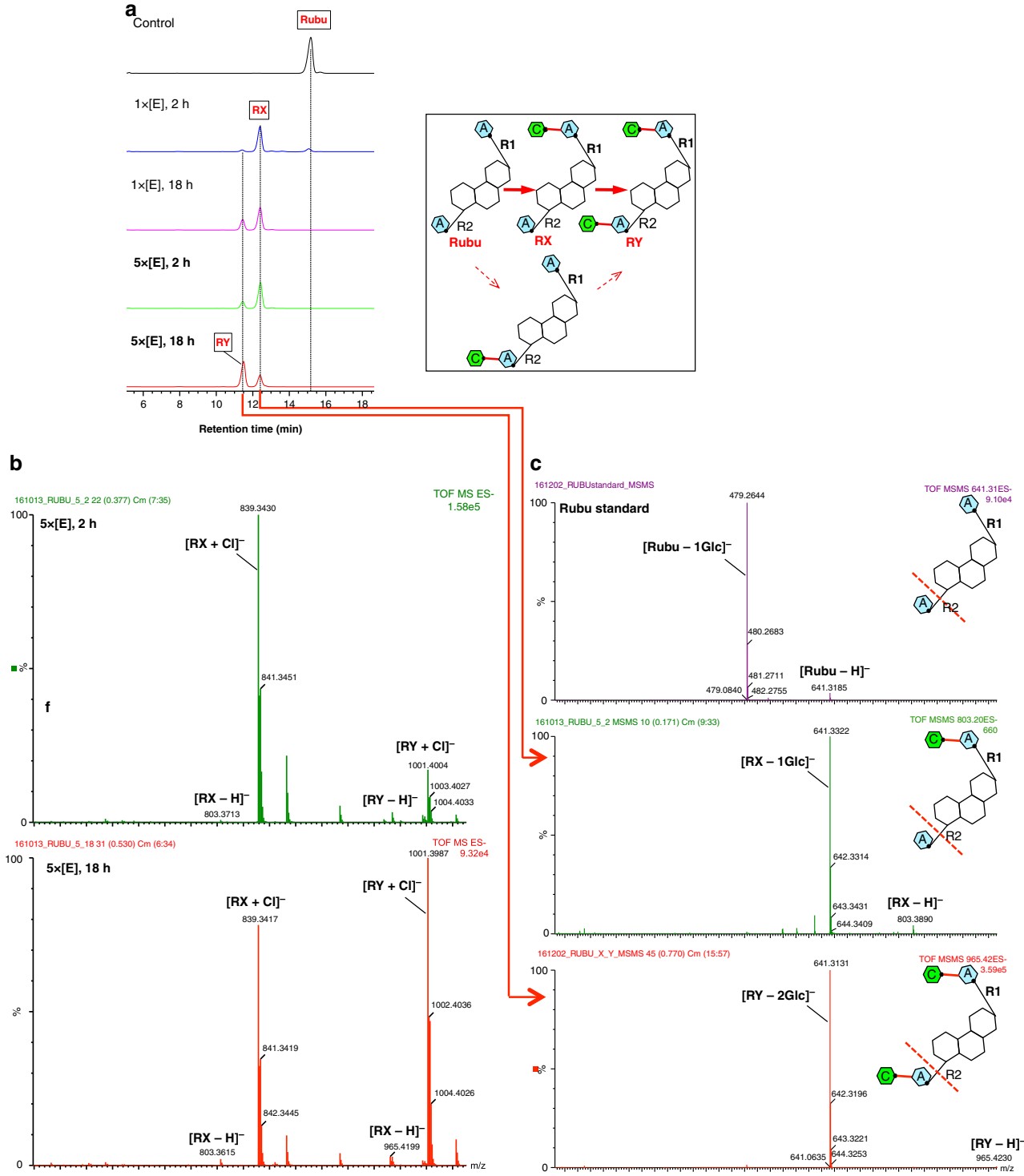

**Fig. 3** Biochemical assays of two sugar transfers to Rubu by UGT76G1. **a** HPLC traces of the reactions of Rubu. The five HPLC traces from top to bottom represent the same reaction conditions as Fig. 2. The first product of Rubu is arbitrarily assigned as RX, and the second one is assigned as RY. The yields of the products are related to the enzyme concentration and the reaction duration. The reactions catalyzed by UGT76G1 are shown in the box. **b** Direct MS of two sugar transfers of Rubu by 0.15 mg ml$^{-1}$ UGT76G1 (5×) for 2 and 18 h. The two main negative ions derived from the products RX and RY are labeled and show the same characteristics in terms of the relative contents of the products as a function of the reaction duration as shown by HPLC (Fig. 3a). **c** MS/MS of the authentic Rubu standard and the collected HPLC peaks of the products RX and RY. The negative ions [Rubu–H]$^-$, [RX–H]$^-$, and [RY–H]$^-$ with m/z at 641.3, 803.4 and 965.4, respectively, were specifically isolated as the parent ions and characterized by MS/MS. The most labile ester bond breaks first, which was consistently indicated by the abundant fragment ions of Rubu, RX, and RY and was used to identify the positions of the added sugars transferred by UGT76G1. The inlet suggests where the ester bond breaks first during MS/MS fragmentation

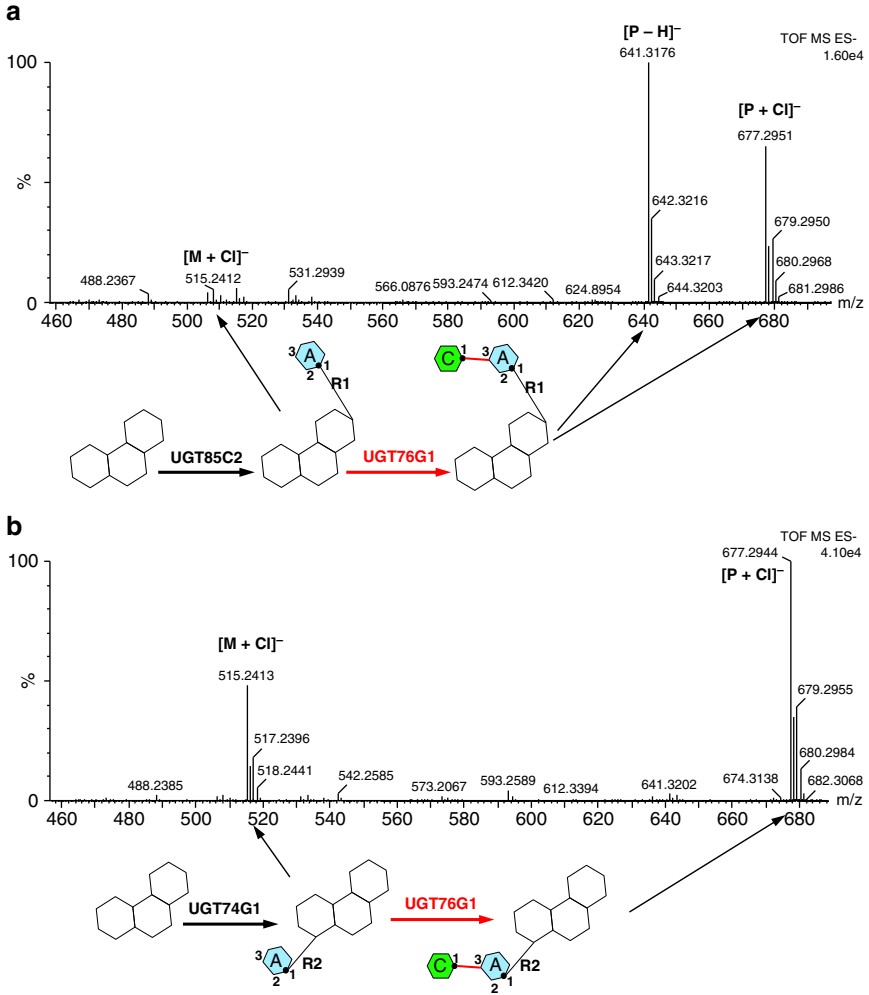

**Fig. 4** The reactions of steviol monoglucosides by UGT76G1. Direct MS of the single sugar transfer at R1 (**a**) and R2 position (**b**) of two self-made steviol monoglucosides. The reaction of the steviol monoglucoside was catalyzed by 0.03 mg ml$^{-1}$ UGT76G1 (1×) for 18 h. The reaction schemes to synthesize steviol monoglucosides and the single sugar transfer catalyzed by UGT76G1 are shown. The main negative ions derived from the products are labeled and related to the chemical compounds in the reaction schemes. The reaction conditions for these two self-made steviol glucosides are identical, and the relative contents of the substrate (M) and the product (P) show the preferred reaction at the R1 position

mass of 641.31 Da, loss of one glucose (−162 Da). This would indicate a single ester-linked glucose at R2, thus glucose transfer of RX occurred at R1 (Fig. 3b, c), consistent with our prediction. [RY−H]$^-$ yields the most abundant product ion with a mass of 641.31 Da, loss of two glucose molecules (−324 Da). This would indicate that RY has an ester-linked disaccharide at the R2 position and arises from the transfer of a glucose at the R2 position of RX (Fig. 3b, c). The lack of standards also precludes confirmation that RX and RY are formed by β (1–3) addition, but given the other compounds, we would expect this to be the case. Support comes from the structure of UGT76G1 with UDP and Rubu (described below), which is consistent with the formation of a β (1–3) linkage at the R1 position.

Our data establish that UGT76G1 can transfer glucose in a β (1–3) manner to an existing glucose A at both the R1 and R2 positions ("ends") of the steviol aglycone. Where possible, both transfers occur, but transfer at R1 precedes transfer at R2. Beyond the requirement for a glucose molecule with a free 3-hydroxyl linked to the aglycone, there seems to be no other strict substrate requirement. To further probe the reaction at the R1 and R2 sites, we biosynthesized two steviol glucosides using either UGT85C2 or UGT74G1 to link a single glucose A to either the R1 or R2 position, respectively (Fig. 1b). Both these steviol glucosides are

modified by UGT76G1 which adds a glucose to either the R1 or R2 end (Fig. 4), once again with R1 addition being faster based on the consumption of the R1 substrate (Supplementary Figs. 7–8).

We measured the steady-state kinetic parameters for a single sugar transfer at the R1 and R2 ends. For those substrates that can accept two sugars (ST, Reb E and Rubu), we focused on the first fast transfer at the R1 position by using short assay times in which consumption of the substrate is less than 10 %; thus, the slow second sugar transfer at the R2 position can be ignored. As expected from our earlier experiments, the $K_m$ and $k_{cat}$ values for R1 glycosylation are more favorable for catalysis than the R2 position in all cases (Table 1). However, the kinetic parameters for transfer to the R2 end of the molecule are well within the range seen for enzymes; in fact, Reb D (R2 transfer) and Rubu (R1 transfer) are similar (within a factor of 3) in terms of $k_{cat}/K_m$ (Table 1). Therefore, we conclude that UGT76G1 is a competent enzyme for catalysis of the addition of β (1–3) glucose to both the R1 and R2 ends of steviol glycosides.

The substrates STB, ST and Reb E (which have a disaccharide) are more rapidly processed than Rubu which has a single glucose. Interestingly, the similar preference for disaccharide over monosaccharide is seen at the R2 end of the molecule (Reb D vs Reb A, Table 1). STB which lacks a sugar at the R2 end and ST

| Table 1 Enzyme kinetic parameters for the single sugar transfer of UGT76G1 | | | | | | | |
|---|---|---|---|---|---|---|---|
| | **UGT76G1** | | | | | | |
| | donor | R1 acceptor | | | | R2 acceptor | |
| | UDPG | STB | ST | Reb E | Rubu | Reb D | Reb A |
| $K_m$ (µM) | 5.6 ± 0.4 | 4.8 ± 0.4 | 25.5 ± 3.4 | 9.2 ± 0.7 | 54.4 ± 4.0 | 94.2 ± 15.5 | 243.4 ± 28.1 |
| $k_{cat}$ (min⁻¹) | 4.1 ± 0.1 | 4.8 ± 0.1 | 41.8 ± 1.4 | 21.3 ± 0.5 | 5.9 ± 0.1 | 3.3 ± 0.2 | 0.12 ± 0.004 |
| $k_{cat}/K_m$ (S⁻¹mM⁻¹) | 12.4 | 16.5 | 27.3 | 38.6 | 1.8 | 0.6 | 0.008 |

Source data are provided as a Source Data file

which has a single glucose show interesting differences, whilst their $k_{cat}/K_m$ values are similar. Compared to ST, STB has a nine-fold decrease in $k_{cat}$ but a five-fold decrease in $K_m$. Reb E which has a disaccharide at both ends, is somewhat intermediate in $k_{cat}$ and $K_m$, but the values of $k_{cat}/K_m$ suggest the best substrate. We conclude that the nature of the R2 position does influence catalysis at R1 but not in a simplistic manner.

**Overall structure of UGT76G1.** We have determined the structures of complexes of both the wild type and H25A mutant of UGT76G1 with UDP, the wild type with UDP and Reb A, and the wild type with UDP and Rubu (Supplementary Table 2). There is one molecule of UGT76G1 in the asymmetric unit and no stable symmetry-related multimer is identified by the PISA web server[21]. Gel filtration suggests UGT76G1 exists as a monomer in solution. The protein structures are all essentially identical within an r.m.s.d. of 0.4–0.9 Å over the Cα atoms. UGT76G1 is a GT-B fold glycosyltransferase[20], which consists of two Rossmann-like fold (β/α/β) domains at the N- and C-termini (Fig. 5a). In the N-terminal domain, seven β-strands (Nβ1–7) form a parallel β-sheet joined by α-helices and loops. A long loop of three α-helices connects Nβ5 and Nβ6. The C-terminal domain has a parallel β-sheet of six β-strands (Cβ1–6), followed by three α-helices (Fig. 5a). The electron density difference maps show clear density for UDP in all structures (Supplementary Fig. 9). UDP sits across the two helices Cα3 and Cα4, contacting three β-strands (Cβ3, Cβ4, and Cβ5), and is perpendicular to the central six-stranded β-sheet of the C-terminal domain. The uridine ring of UDP π-stacks against Trp 338 and forms two hydrogen bonds to the backbone (amide and carbonyl) of Val 339. The O2 and O3 hydroxyl groups of the ribose ring form a bidentate hydrogen bond with Glu 364, and a hydrogen bond forms between O3 hydroxyl and the side group amide of Asn 27. Two of the α-phosphate oxygens are bound by the main chain amides and side chains of Asn 360 and Ser 361. The pyrophosphate oxygen and two oxygen atoms of β-phosphate are linked to the NE2 of the imidazole ring of His 356 and to the amide and hydroxyl group of Ser 283, which is in the flexible loop between the Cβ1-strand and Cα1-helix (Fig. 5b and Supplementary Fig. 10). Upon Reb A or Rubu binding, the β-phosphate shows a slight shift, but the rest of UDP remains unchanged in our structures. The interactions with UDP observed here are conserved in other GT-B fold glycosyl-transferases[22,23] (Supplementary Fig. 10a).

In both ternary complexes (with UDP and Reb A and with UDP and Rubu), two steviol glucosides are unambiguously located in the electron density (Supplementary Fig. 9). One molecule is bound next to UDP at the active site (Fig. 5a). The other molecule is located at the edge of the parallel β-sheet of the N-terminal Rossmann domain. Since this second binding site is remote from UDP, we conclude that this site is an artifact of the high concentrations of the steviol glucosides used in crystallization and do not discuss it further. The trisaccharide ABC of Reb A and glucose A of Rubu at the R1 position are clearly

defined; however, the glycones at the R2 position are not located in the electron density, although the R2 groups are ordered and defined in the artifact binding site (Supplementary Fig. 9). The incubation of Reb A or Rubu with UGT76G1 and UDP in the crystallization buffer for one week (the crystallization duration) did not result in any detectable hydrolysis products. Therefore, we concluded that the failure to observe the R2 group in Reb A and Rubu was attributable not to chemical degradation but to disorder or flexibility consistent with the lack of recognition. Although the R2 glycone clearly alters the kinetics of the substrate for transfer at R1, we are unable to offer a structural rationale for this behavior.

**Structural basis of recognition by UGT76G1.** The UDP/Reb A structure can be thought of as representing the product complex formed by sugar transfer to the R1 position of ST (Fig. 5c and Supplementary Fig. 12). The UDP/Rubu structure mimics a substrate complex (Fig. 5d and Supplementary Fig. 12). For ease of discussion, the three glucose rings of R1 glycone in Reb A are denoted by glucose AR1, BR1, and CR1 (Fig. 1a). The position of the glucose AR1 is critical for regioselective catalysis as its 3-hydroxyl group attacks UDPG during the sugar transfer. Interestingly, glucose AR1 lacks extensive hydrogen bond recognition and is surrounded by Leu 379, Phe 22 and Ile 90 (hydrophobic patches) with only a single hydrogen bond between the 3-hydroxyl oxygen and the side chain of His 25 (Fig. 5c). The H25N mutation completely eliminates the activity on both R1 and R2 sites as judged by MS. A trace of the product can be detected by MS in the R1 reaction of H25A, the $k_{cat}$ value of such a reaction is $10^{-5}$ of that of the wild-type value in the steady-state enzyme assays (Supplementary Fig. 11). The inactivity of the His 25 mutation indicates that the same catalytic machinery is responsible for the sugar transfer at both the R1 and R2 ends. We propose that His 25 is the general base, which deprotonates the 3-hydroxyl of the accepting glucose A to activate it as a nucleophile (Fig. 5e). The nucleophile attacks UDPG to form the transition state, followed by a $S_N2$ reaction with UDP as the leaving group[24]. Asp 124 forms a dyad with His 25, suggesting it could play an important catalytic role in relaying protons off and on His 25. We were unable to test this as both the D124N and D124A mutants were insoluble. The structure of H25A shows that a cluster of ordered water molecules occupy the position of His 25 and form hydrogen bonds to Asp 124, which might act as a much worse proton relay for the trace activity (Supplementary Fig. 11). The H25N mutation prevents a similar water molecule arrangement.

In contrast to glucose AR1, the 2-, 3-, and 4-hydroxyl groups of the glucose BR1 (which is β (1–2)-linked to AR1) form a network of interactions with His 25, Ser 147 and His 155. This network is consistent with the improved $K_m$ and $k_{cat}$ values for those steviol substrates (ST, STB, Reb E) that have disaccharide acceptor. Glucose CR1, β (1–3)-linked to AR1, derives from the transfer from UDPG, and it forms extensive interactions with the protein through every hydroxyl except the 2-hydroxyl. The

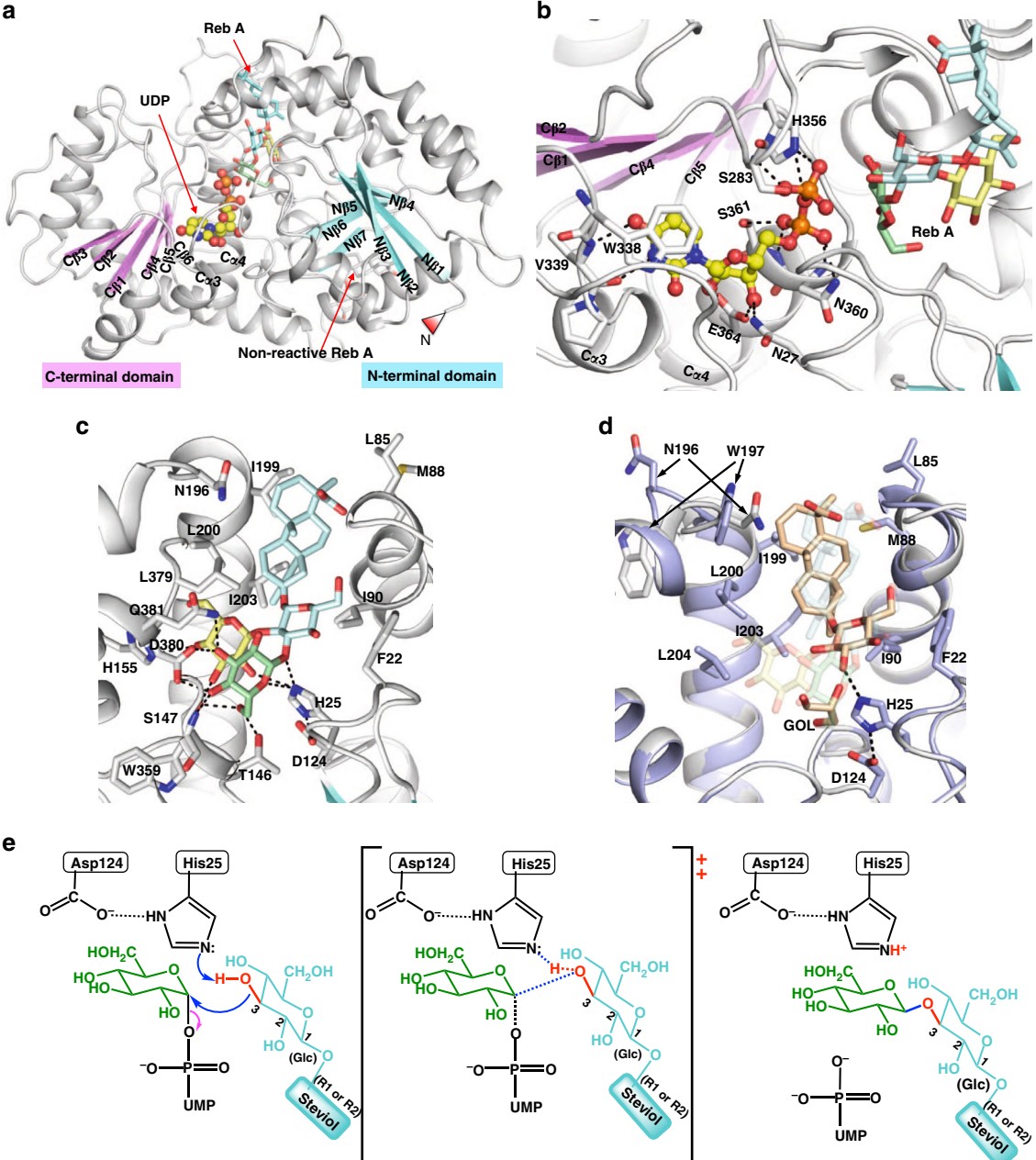

**Fig. 5** The structure of UGT76G1 and analysis of the active site. **a** The overall structure of UGT76G1. The structure is shown in cartoon representation. The β-sheet of the N-terminus is shown in cyan and that of the C-terminus in pink. The bound UDP is shown in stick-ball mode with carbons colored yellow, oxygen red and nitrogen blue. Two molecules of the steviol compound Reb A are shown in stick representation with oxygen colored red, carbons of the nonreactive Reb A colored white, carbons of the reactive Reb A colored differently, carbons of steviol aglycone and the glucose A in cyan, the glucose B in light yellow and the incoming glucose C in pale green. **b** The binding pocket of UDP in the presence of Reb A with atoms colored as shown in Fig. 5a. The hydrogen bonds are shown by dashed lines. **c** The binding pocket of the reactive Reb A. **d** Comparison of the ternary complex structures with Rubu and Reb A. Two ternary complex structures are shown in cartoon representation with the proteins in light blue (Rubu) and white (Reb A). The residues associated with Rubu are shown in stick representation with carbons colored light blue, oxygen red and nitrogen blue. Two residues, Asn 196 and Trp 197, in the structure of Reb A are shown in stick representation with carbons colored white, which highlight the flexible loop between Ser 195 and Lys 201. Rubu and a glycerol molecule (GOL) in the active site are shown in stick with carbons colored gold, while Reb A is shown in a transparent stick representation after overlapping their ternary complex structures. The glycerol molecule occupies a similar position of the incoming glucose C of Reb A. **e** The $S_N2$ mechanism of UGT76G1 catalysis. His 25, with the aid of Asp 124, specifically deprotonates the 3-hydroxyl of the glucose A and makes it a nucleophile to attack the sugar donor UDPG. The transition state of the $S_N2$ mechanism is shown in the brackets

3- and 4-hydroxyl groups together form a bidentate hydrogen bond with the carboxylate group of Asp 380, in addition to multiple other hydrogen bonds and van der Waals interactions. We were unable to obtain a UDPG complex structure. However, the oxygen of the β-phosphate of UDP and the C1 carbon of the glucose CR1 are separated in the UDP/Reb A complex by only 3.2 Å and aligned such that it is trivial to make a simple model of UDPG binding by taking the glucose CR1 as the glucose of sugar donor UDPG. This simple model is reasonable, as glucose CR1 maintains the same extensive interactions and overlaps

approximately with the glucose of the experimentally determined UDPG or UDPG analog in related glycosyltransferase structures (PDB codes 2acw and 2c1z)[22,23] (Supplementary Fig. 10c).

In the Rubu/UDP complex, the glucose A at the R1 end corresponds with the glucose we denoted AR1 in the Reb A complex. Once again, this sugar lacks a hydrogen bond network of recognition, and we observe a small shift of the glucose molecule when compared to Reb A (Fig. 5d). The 3-hydroxyl group of glucose A of Rubu still forms a hydrogen bond to the NE2 nitrogen of the catalytic residue His 25, but the geometry is different (Fig. 5c, d), correlating with the lower activity of Rubu. We attribute this shift to the absence of glucose B, the β (1–2)-linked glucose (denoted BR1 in Reb A). We conclude that the presence of the β (1–2)-linked glucose B anchors the substrate, enhancing catalysis. In the Rubu complex, a glycerol molecule occupies the same position as glucose CR1 of Reb A, which can represent the glucose transferred from UDPG.

In both the Reb A and Rubu complexes, the diterpenoid steviol aglycone ring sits in a hydrophobic pocket formed by Leu 85, Met 88, Ile 90, Asn 196, Ile 199, Leu 200, and Ile 203 (Fig. 5c, d). The diterpenoid aglycone rings do not overlap and are offset by a 30° rotation, which is attributed to the lack of a specific hydrogen bond recognition. Consequently, there are shifts in the protein structure between these two complexes, notably the flexible loop between Ser 195 and Lys 201.

Our assays identified the minimal substrate to possess the steviol aglycone ring and glucose A at either the R1 or R2 site. It was interesting to note that it is these two essential components that are predominantly recognized by hydrophobic interactions (simultaneous van der Waals interactions) with almost no hydrogen bonds. As a result, each component sits in a cavity, and we see in the crystal structures that the different steviol glucoside molecules (Reb A and Rubu), which possess the minimal substrate, adopt different positions for both the glucose and the steviol aglycone ring due to the lack of a typical hydrogen bond network for the sugar substrate. However, the origin of the specificity is clear, in which two substrates, UDP-glucose and a glucose with a hydrophobic steviol aglycone ring, are oriented in such a way that the 3-hydroxyl of the glucose is activated by His25 and points toward the C1 of UDP-glucose (Fig. 6a, c).

**The role of the steviol diterpenoid aglycone.** Since the enzyme processes at both the R1 and R2 ends using the same catalytic machinery, we constructed a model that placed the R2 end at the active site, essentially flipping the molecule. To position the R2 glycone at the active site, we used the nonreactive Reb A molecule that was located far from the active site with an ordered glycone at the R2 end. We guided the model to position the 3-hydroxyl of the glucose appropriately for catalysis by His 25. Finally, the "flipped" steviol aglycone was positioned using the Reb A and Rubu structures. The geometry of this crude model complex was then idealized by Refmac[25,26]. The R1 and R2 ends of the original structure and the "flipped" model almost perfectly superimpose despite the steviol aglycone lacking any such internal two-fold symmetry (Supplementary Fig. 13). The minimized flipped model retains the 3-hydroxyl of the R2 glucose group for activation by His 25 (Fig. 6b, c). Comparing the interactions with the steviol aglycone in the flipped model and the original complex reveals significant differences but no impossible clashes. The minimized flipped model rationalizes the ability of the enzyme to process the R2 end of the molecule (Fig. 6b, c). We presume that the altered interactions in this flipped form are responsible for the higher $K_m$ and lower $k_{cat}$ values seen for substrates processed at the R2 end.

Since we observed considerable flexibility in the steviol ring aglycone location in the crystal structures, and our flipped model

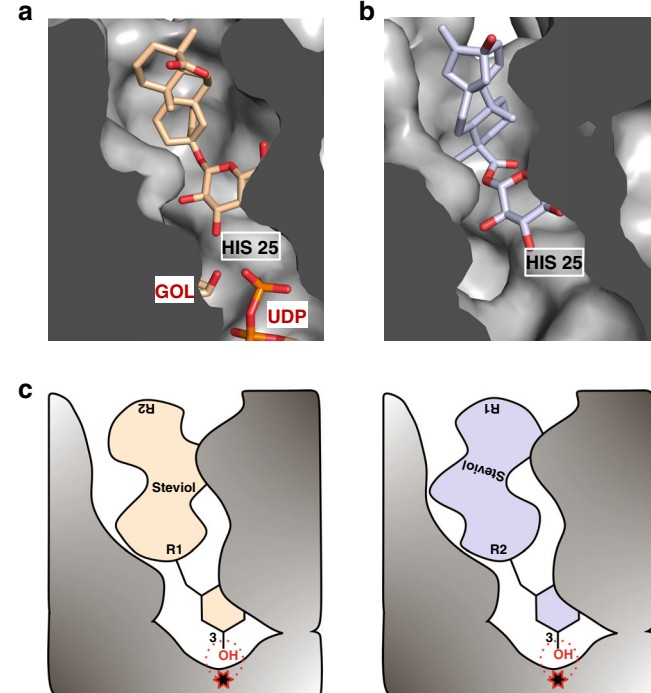

**Fig. 6** The two binding modes of the steviol substrates for the sugar transfers. **a** The normal binding mode for the R1 reaction. The binding mode for the R1 reaction is represented by the ternary complex structure with Rubu, in which the positions of Rubu, GOL, UDP and the catalytic residue His 25 are shown in the cross-section of the binding pocket. **b** The flipped binding model for the R2 reaction. The flipped binding mode for R2 reaction is modeled in the ternary complex structure with Reb A, in which the flipped steviol glucoside fragment (with carbons colored light purple) stretches the 3-hydroxyl of the glucose AR2 toward the catalytic site without any obvious clash in the binding pocket. **c** The cartoon representation of the normal and flipped binding modes. The catalytic residue His 25 and its reactivation region are represented by a red star and red dots, respectively, toward which the 3-hydroxyl group of either the R1 or R2 position would be presented

does not clash, we hypothesized that it might be possible to substitute steviol diterpenoid aglycone with another hydrophobic structure. We chose 4-nitrophenyl β-D-glucopyranoside (Fig. 7) and incubated it with UGT76G1 and UDPG. We detected a new product with a mass increase of 162 Da (equivalent to glucose) relative to 4-nitrophenyl β-D-glucopyranoside. The amount of this new product varied with both time and the amount of enzyme in the assay (Supplementary Fig. 14a). MS/MS of the product ion identified five characteristic ion fragments of the disaccharide (Supplementary Fig. 14b), indicating that the enzyme glycosylates at the glucose of 4-nitrophenyl β-D-glucopyranoside. We conclude that the enzyme requires only a relatively flat hydrophobic aglycone for a sugar substrate.

## Discussion

The enzyme UGT76G1 shows an unusual degree of plasticity in substrate recognition; it is able to transfer a glucose to two different ends of some steviol glycosides despite having a single active site. Since the steviol aglycone has no internal symmetry, the enzyme is effectively able to recognize two different substrates at the same active site (in fact, one substrate in two different binding orientations). This is relatively rare since enzymes are often highly specific for their substrates or the binding orientation.

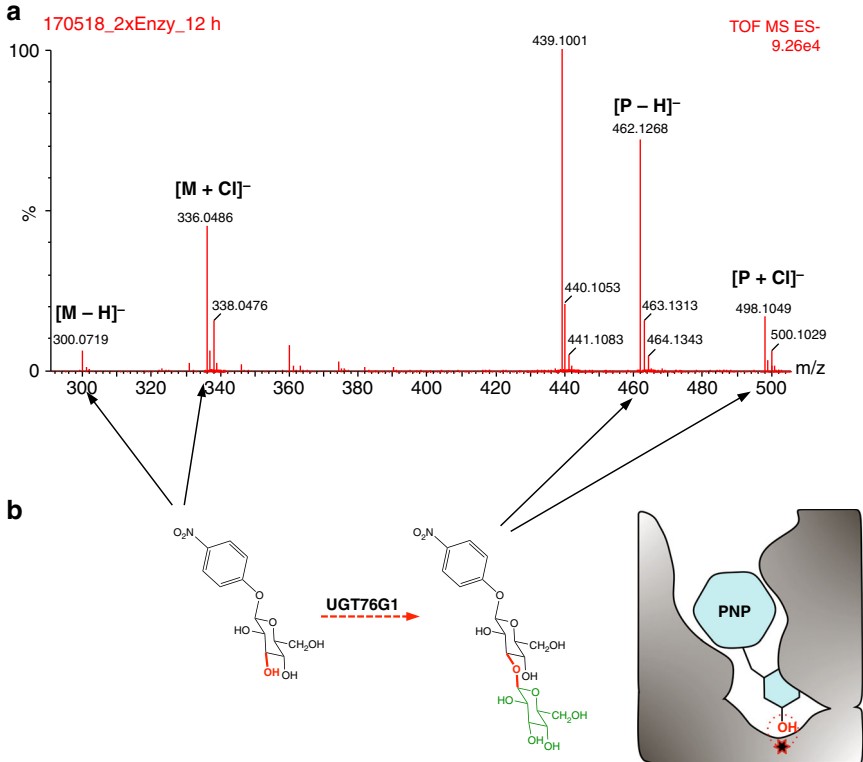

**Fig. 7** Replacement of the steviol aglycone ring of the substrate by another hydrophobic group. **a** Direct MS of the reaction of 4-nitrophenyl β-D-glucopyranoside by UGT76G1. The nitrophenyl ring shows a similar role to give the reactivity to the glycone unit. The reaction of 4-nitrophenyl β-D-glucopyranoside was catalyzed by 0.25 mg ml$^{-1}$ UGT76G1 for 12 h. The two main negative ions derived from the substrate (M) and product (P) are labeled. **b** The slow reaction scheme and the binding mode of 4-nitrophenyl β-D-glucopyranoside by UGT76G1

For its natural substrates, the enzyme clearly has a preference for the R1 end of the molecule, and catalysis is improved by having a disaccharide rather than a monosaccharide. However, kinetic analysis shows the enzyme is competent to catalyze glycosylation at the R2 end, showing the same acceleration if a disaccharide rather than a monosaccharide is present. We made two simple steviol glucosides, one with a glucose at R1 and one with a glucose at R2. As expected, the R1 variant was processed rapidly, but the R2 variant was also processed, albeit more slowly. This experiment supports three related points: firstly, the essential components to become a substrate are a hydrophobic ring (like the steviol diterpenoid aglycone) and a glucose with a free 3-hydroxyl (the accepting sugar) attached to that ring. Secondly, the apparent reaction order from the R1 to the R2 site is a function of kinetics, and there is no requirement for modification of the R1 site for the R2 site to become a substrate. Thirdly, the ability to modify a glucose at R2 is a function of the presence only of the steviol aglycone moiety. A simple model shows that the enzyme is able to bind the steviol aglycone in a flipped orientation that would allow transfer to glucose attached to the R2 end of the steviol aglycone without any obvious clashes. Economically, this is important since it is the presence of sugars at the R2 end transferred by UGT76G1 that gives the highly desired sweetness without bitterness.

Analysis of the crystal structures reveals that in the acceptor substrate, neither the sugar and nor the aglycone participates in a network of hydrogen bonds. Rather both moieties are held in place by hydrophobic (van der Waals) interactions. While such interactions give strength to binding and have a distance dependence, they do not have a well-defined three-dimensional dependence. This is in contrast to the geometrical requirement of the hydrogen bonds normally seen for acceptor substrate

recognition in glycosyltransferases[27,28]. Comparison of two complex crystal structures of UGT76G1 with Reb A and with Rubu shows that the aglycone portions are bound with significantly different orientations. Such "wobbling" or flexibility in binding is consistent with the hydrophobic interactions used by the enzyme to recognize the relatively tubular hydrophobic steviol aglycone molecule. Although hydrophobic interactions associated with the steviol aglycone are essential for binding, we suggest they lie at the heart of substrate plasticity. It is the hydrophobic interactions made by the steviol aglycone in both the normal and the flipped form that allows the enzyme to operate on both ends. Such a strategy would be inconceivable if recognition was based on hydrogen bonding. As an analogy like velcro fastening, the use of hydrophobic interactions means that a precise orientation of the components is not required for function.

The use of an attached recognition handle outside of the reacting portion of a sugar molecule is common in biology and most clearly illustrated in sugar nucleotide chemistry[29]. The nucleotide serves as a handle to anchor substrates using both hydrogen bonds and van der Waals interactions, and these enzymes are highly specific to their substrates. In the case of UGT76G1, nature has created an enzyme that can process different substrates by relying on hydrophobic recognition, which allows the synthesis of different glycans with the same shared set of enzymes. This redundancy reduces the number of enzymes required to make a diverse set of molecules. There are other glycosyltransferases that catalyze sugar transfer to hydrophobic aglycones[23,30,31], and it will be interesting to see if they possess similar flexibility. Utilizing purely hydrophobic recognition handles to broaden the scope of glycosyltransferases points toward new avenues in biotechnology. Since such recognition is spatially less constrained than hydrogen bonding, engineering of the

protein or the handle may be much easier than in the case of glycosyltransferases that rely on hydrogen bonding. In the specific case of steviol glucosides, these data lay the foundation for the deployment of the enzyme to perform the conversion to the healthy zero-calorie sugar substitutes Reb D and Reb M for mass production.

## Methods

**Protein expression and purification**. The encoding region of UGT76G1 of *S. rebaudiana* was de novo synthesized and subcloned into the expression vector pET21a between *Nde I* and *Xho I* (Supplementary Table 1) so that the expressed UGT76G1 contained a hexa-histidine tag at the C-terminus. The constructed plasmid pET21-UGT76G1 was transformed into BL21 (DE3) (Novagen) for overexpression. After induction with 0.5 mM IPTG for 4 h at 37 °C, the *E. coli* cells were harvested by centrifugation and then lysed by sonication in 20 mM Tris-HCl buffer pH 7.8, 300 mM NaCl and 20 mM imidazole. The His-tagged UGT76G1 was eluted from a Ni$^{2+}$ HisTrap affinity column (GE Life Sciences) by 20 mM Tris-HCl buffer pH 7.8, 300 mM NaCl and 250 mM imidazole, followed by a final gel filtration step in 10 mM HEPES-NaOH buffer pH 7.2, 150 mM NaCl, and 2 mM DTT. The protein was finally concentrated to 20 mg ml$^{-1}$ and stored at −80 °C.

**Structural biology**. The protein (20 mg ml$^{-1}$) was mixed with 1 mM UDP and the steviol glucoside compounds, and crystallization trials were set up using the sitting-drop vapor-diffusion method at 20 °C. A number of crystallization conditions were identified from commercial sparse matrix screens. Linear optimization of the crystallization conditions and subsequent preliminary X-ray analysis identified diffraction-quality crystals. The best crystals, judged by visual inspection, were obtained under the crystallization condition consisting of 0.1 M sodium citrate buffer at pH 5.4 and 20% PEG 4000. The crystals were cryoprotected for data collection by supplementing the above crystallization conditions with 20% glycerol.

X-ray diffraction data were collected at beamline BL19U1 at Shanghai Synchrotron Radiation Facility (SSRF, National Center for Protein Science Shanghai, Institute of Biochemistry and Cell Biology, Chinese Academy of Sciences, P. sR. China) using a Pilatus detector at a wavelength of 0.97853 Å. The data were indexed, integrated and scaled using the DIALS package[32]. The structures were solved by the molecular replacement method with the structure of UGT85H2[33] (PDB code 2PQ6, 29% identity) as the search model by using Phaser. Manual model building and subsequent refinement were performed using Coot[34] and Refmac[25,26]. The restraint libraries for Reb A and Rubu were produced by JLigand[35] and idealized by Refmac. The final data collection and refinement statistics are summarized in Supplementary Table 2.

**Biochemical assays and MS analyses**. The simple in vitro glucosyltransferase assays were performed by high-resolution MS or additional HPLC in triplicate. 0.3 mM of each Substrate, including STB, ST, Reb A, Reb E, Rubu, and Reb D, was assayed in the reaction buffer (20 mM Tris-HCl buffer pH 7.2, 1 mM UDPG). The reactions were initiated by adding 0.03 or 0.15 mg ml$^{-1}$ recombinant UGT76G1 and incubated at 25 °C for 2–18 h (unless otherwise indicated). The reactions were terminated and extracted by adding an equal volume of water-saturated 1-butanol. After centrifugation at $17,000 \times g$ for 10 min, the upper butanol layer was collected and subjected to direct analysis by high-resolution MS in the negative ion mode. Additionally, the extracted products were resolved using a Luna 5 μm C18 100 Å HPLC column (Phenomenex) and were eluted with an increasing acetonitrile gradient (27% acetonitrile for 2 min, 27–60% in 20 min, 95% for 3 min and back to a constant 27% acetonitrile for 2 min). All products were identified by comparison to authentic steviol glucoside standards (ChromaDex, USA) if commercially available. The product peaks of interest from HPLC were concentrated to 10 μl using a SpeedVac (Thermo Savant) and further verified by MS or MS/MS analyses.

MS analysis was carried out on a Q-TOF Premier™ with ESI Source in negative ion mode (Waters, Milford, MA). The Q-TOF Premier™ is a hybrid orthogonal acceleration time of flight mass spectrometer that enables the automated exact mass measurement of precursor and fragment ions to yield the highest confidence in structural elucidation and databank search results. The mass spectra were recorded using full scan mode over a mass range of m/z 100–1600 in negative ion mode. The optimized ESI source parameters were set as follows: capillary voltage at −2.8 kV; sampling cone voltage at 40 V; extractor voltage at 3.5 V; source temperature at 90 °C; desolvation temperature at 250 °C and desolvation gas (N$_2$) at a flow rate of 120 l h$^{-1}$. The Q-TOF-MS/MS experiments were performed by setting the quadrupole to specifically isolate and characterize the ions of interest. The collision cell parameters for the Q-TOF-MS/MS experiments were as follows: collision gas (argon) flow rate, 0.45 l h$^{-1}$; collision energy, 20–50 eV. The mass and MS/MS acquisition rates were both set to 1.0 s, with a 0.02 s interscan delay.

To examine the minimal essential components of the substrate, two steviol glucosides, which are not commercially available, were synthesized by UGT85C2 or UGT74G1, respectively according to the reaction scheme shown in Fig. 1b. UGT85C2 and UGT74G1 were expressed and purified as UGT76G1 in this study. Pure steviol aglycone (ChromaDex, USA) at 0.3 mM was converted by 0.03 mg ml$^{-1}$ UGT85C2 or UGT74G1 for 2 h under the same reaction condition of UGT76G1. The respective product of the conversions had a single glucose at the R1 or R2 position of the diterpenoid steviol aglycone, which was further tested the catalysis by UGT76G1 as other steviol substrate standards.

**Steady-state enzyme kinetic assays**. After confirming the reactivity of the substrates by MS or HPLC analyses, steady-state kinetic assays of those substrates were performed by using the UDP-Glo™ Glycosyltransferase Assay Kit (Promega). The UDP released from the reaction catalyzed by UGT76G1 is converted to ATP, which is further quantified by using a luciferase/luciferin reaction. The luminescence intensity linearly correlates to the concentration of UDP, which reflects the glycosyltransferase reaction rate in the time course. The reaction was initialized by the enzyme, and four individual aliquots were taken every 30 s or 1 min (based on the reaction velocity in the pilot experiment) to mix with an equal volume of the UDP detection reagent, which terminates the reaction and quantifies the released UDP. The proper concentration of purified UGT76G1 was used to maintain the consumption of both steviol substrate and UDPG is <10% of the initial concentration during the assay duration. For those steviol substrates that have two sugar transfers, at both the R1 and R2 sites, we measured only the preferred single sugar transfer at the R1 site by choosing a short reaction interval in which the substrate is consumed by less than 10 % of the original concentration, and the reaction at the R2 position can be neglected. A standard linear plot between the luminescence intensity and UDP concentrations was drawn for every measurement ($R^2 > 0.99$), and the blank turnover of UDPG to UDP in the absence of the steviol substrate was subtracted from each measurement. All of the assays were performed in triplicate. The steady-state kinetic parameters of the steviol substrate were determined by nonlinear fitting according to the Michaelis–Menten equation when UDPG is at 100 μM, which is more than 10 times its $K_m$ and vice versa (the $K_m$ of UDPG is measured at 5.6±0.4 μM when the steviol substrate STB is at a concentration of 50 μM). The source data are provided as a Source Data file.

**Reporting Summary**. Further information on research design is available in the Nature Research Reporting Summary linked to this article.

## Data availability

The coordinates and structure factors have been deposited in the Protein Data Bank under accession codes 6INF, 6ING, 6INH, 6INI. All relevant data are included in the paper, the Supplementary Information file or the Source data file. The source data underlying the kinetic parameters in Table 1 are provided as a Source Data file. Other data are available from the corresponding authors on reasonable request.

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

## Acknowledgements
We thank the staff of the BL17U1 beamline and BL19U1 at the Shanghai Synchrotron Radiation Facility, Zhangjiang Lab, for assistance during data collection. This work was funded by grants to X.Z. from the National Natural Science Foundation of China (31771910, 21534008), National Key Research and Development Program of China (2018YFC1002803), the Research Program of Chengdu Science and Technology (2015-HM01–00504-SF) and Sichuan Province Thousand Talents Scheme in China. J.H.N. is supported by the Wellcome Trust (100209/Z/12/Z) and the Chinese National Thousand Talents Program.

## Author contributions
X.Z., J.H.N. designed experiments, interpreted data and wrote the manuscript; X.Z. performed the crystallization, HPLC and mass spectrum analyses, Q.L., G.C., J.L., W.C. and Y.W. contributed to materials, data analysis and manuscript, T.Y., D.K., J.Z., W.Y., M.T., J.J. and Y.Z. performed the biochemical characterization, mass spectrum analyses, and crystal diffraction data collection; X.Z and J.H.N solved and analyzed the X-ray crystal structures. All authors approved the final manuscript.

## Additional information

**Competing Interests:** The authors declare no competing interests.

