## [Peer Review File · Nature Communications]

Reviewers' comments:

Reviewer #1 (Remarks to the Author):

The manuscript with the title "Hydrophobic recognition allows the glycosyltransferase UGT76G1 to catalyse the substrate in two orientations but retain regiospecificity" by Yang et al. describes the structural and functional characterisation of an glycosyltransferase from *S. rebaudiana* involved in the production of the sweeteners RebM and RebD, which are preferred due to their specific characteristics in receptor activation.

The authors characterise the enzyme using activity assays, binding studies and x-ray crystallography to unravel the unique characteristics of the enzyme in respect of substrate specificity and activity.

Overall the manuscript is very interesting and the body of work is carried out well, however at times it looks as the manuscript is put together a bit sloppy. It would benefit from an additional round of careful reading.

Specific comments to the manuscript are outlined below.

Introduction:

Would be highly beneficial \diamond is highly beneficial

Glycosylic bonds \diamond glycosidic bonds

Microbial production of Reb M and Reb D present potential solutions

It is mentioned that UGT76G1 is the only known enzyme to perform beta 1-3 glycosylation in *S. rebaudiana*. Are there other enzymes for the beta 1,2 addition?

Results

Though most GT'S are metal dependent, there are well documented examples, where this is not the case.

....and high resolution mass spectrometry (MS) rather than spectroscopy and also the "analyses" can be removed

Which other substrates besides the mentioned steviosides are glycosylated? One artificial one is mentioned, but the sentence implies there are more. Are these other steviosides ore something completely differently? The mentioned one have been used in the paper.

Very interesting is the analysis of the order of the reaction. It seems quite obvious that the R1 site is preferred, but is there really a very strict order? There is unfortunately now stevioside with just the R2 site modified without sugars at the R1 site. Also might it be that the RebE R1 mod and R2 mod run in HPLC at the same elution time? So a minor product of R2 modified first might be not really detectable over the majority of R1 modified first product.

"There is only one molecule in the asymmetric unit since no stalbe multimer is identified using PISA. "

I'm afraid, but this statement is wrong The oligomeric state and the content of the asymmetric unit are two different and unrelated things. There can be monomers in the au but the molecule is an oligomer in solution and in the crystal, if symmetry related monomers are taken into account. Rmsd \diamond r.m.s.d.

Central six-strand beta - \diamond central six-stranded

Generally speaking in the conformation observed, Reb A might be a product structure, but in principle it can undergo a second glycosylation at R2. The sentence should be rephrased to make that qualification.

One should be careful with a statement of the transition state modelling. There is still controversy about the exact mechanism and therefore concomitant transition state for quite some GT's despite extensive trials to clarify the mechanism.

From Figure 3 I cannot see or understand the claim that in Rubu the β 2 glucose would Clahs with His25 or Leu 204. That entire subparagraph is hard to understand and it is not clear what the authors want to say with that paragraph.

The authors said they cannot get binding curves with UDPG due to reaction heat. That is correct, but allows at the same time to get an Km and kcat value from the enzymatic reaction if the experiment is done correct.

Contributing entropic and enthalpic components to different states is tricky and might be not justified here Especially the interference of bonding statement is very confusing and it is not clear what the authors want to argue here.

For the bi-bi mechanism to work the authors say UDPG binds first and the steviol second.

Therefore and in line with the conserved binding mode of UDP(G) and analogs between different families it seems unlikely that the binding of steviol will contribute significantly to changes in the binding of UDP by movements not related to residues involved in binding of the UDP(G)

It is noteworthy that despite an expected stoichiometric 1 to1 binding and a μM binding constant, the N value is often significantly below 1. That indicates problems in the model or the concentrations of the protein or ligand. Furthermore despite reasonably good binding constant the lower plateau is not at all visible in any of the binding isotherms

The paragraph arguing about binding of the substrates/ products and UDP and kinetically favoured bindings is again very hard to read and understand. I think the product with multiple hydrogen bonds binds readily. Especially taking into account that there is also a rearrangement of loop regions of the protein. I what way the beta 2 glucose does contribute to the overall reaction is debateable and needs to be further clarified.

Reb I shows no binding with UDPG this is presumably for steric reasons, but Reb I is a product and should anyway show very weak binding to an ! active complex. A titration of Reb I in UDP enzyme might be interesting If it is purely steric than Reb I should have quite good affinity to the enzyme Reb B with UDP should give similar Kd values as ST STB and RebE. That would be a good and necessary control. ReB B doesn't contribute to a flipped binding mode. It simply binds that way due to steric clashes with UDPG according to the authors interpretation.

Discussion:

...include the hydrophobic stevil ring and a glucose

Overwhelm the importance? What is meant with that phrase?

Reb B contribution: I wonder if the steviol binds on its own similar good as Reb B That would be a good experiment to verify the claim and show unequivocally that this is indeed the case.

The whole paragraph discussing binding modes in the light of potential transition states and interactions explaining the reaction order and potential activation barrier is highly speculative and with the currently available data hardly justifiable. That should be rewritten and be founded more on the actual data.

The idea presented to uses hydrophobic handles is a very interesting concept and it will be indeed very interesting to see if other enzymes does work in the same way.

Methods:

Data collection parameters indicate very good statistics. It is unclear why the data were cut off at the values chosen here. The CC1/2 I/sig I all indicate that the crystals diffracted to far higher resolution.

Figures:

Figure 3:

I would suggest to go a bit more off white in Fig 3a to enhance the contrast for the backbone

Figure 3c and d would benefit from being stereo due to their complexity

Figure 5a)

It is hard to give suggestions how to improve this figure. It is hard to see what belongs to what.

One option would be to have the scheme (chemdraw) of the two steviol compounds next to it to help to navigate through that overlay.

Figure 5c)

Thought the scheme itself is nice, it suggests a considerable structural plasticity (domain movement), which does not take place. There is a rearrangement of a loop but not major movements. At least not described in the manuscript.

Supplementary

Fig S1. This figure seems unnecessary. The activity assay is much clearer in regard of the reactivity of the enzyme in respect to metal ion content. Taken into account would be only

important to coordinate the phosphate and neutralise the charge.

Fig S2: Some of the labels are hard to read as they overlap with the peaks. It might help to move them a bit.

Fig. 9Sb could be also worth doing it in stereo.

Generally often analyses is used, instead of analysis. For example

Fig S12: It is MS and Ms/Ms analysis of the reactions.....

Figures S10 The H25A structure inlet overlays with the MS spectrum below. Furthermore I can't find an explanation of the inlet in the legend of the figure.

The waters are discussed in the main text and it is important for the discussion. This part should go in one of the main figures to explain better how the waters compensate for the lack of the His side chain.

Reviewer #2 (Remarks to the Author):

This paper reports the x-ray structure of UGGT6G1 (and complexes) a glucosyltransferase involved in the biosynthesis of steviol glycosides, natural products that are used as sugar substitutes. It also reports experiments aimed at characterizing the molecular basis for the observed substrate binding and catalytic properties shown by this enzyme.

Steviol possesses a diterpenoid backbone which is glycosylated (up to trisaccharides) at both ends (ie. R1 and R2 end) to generate the various naturally occurring steviol glycosides. The main conclusion of the paper is that UGGT6G1 recognizes the steviol aglycone through apolar interactions that are not dependent on a unique structural mode of interaction. In this way, the enzyme can glucosylate the saccharide moiety at the R1 and R2 ends by binding to the steviol backbone in two different ways.

The x-ray structures have been determined at high resolution and based on the data collection and refinement statistics they seem to be of high quality. The ITC and rate measurements also seem to be sound.

Although the structures are novel and the system is an interesting one, the paper is lacking at a number of levels. Perhaps the greatest flaw is the lack of enzyme kinetic analysis. Any attempt to rationalize specificity differences such as the R1/R2 end preference and the role played by the β 1,2-linked glucose moiety require a measure of both K_m and k_{cat} for the various substrates.

In a number of places the material presented and/or discussed is highly speculative, poorly communicated and/or not central to the thesis of the paper. The discussion of the differences in enthalpy/entropy of binding shown by UDP and UDP-glucose provide an example. At the end of that section the sentence beginning with "The different energy profiles between" is incomprehensible. With regard to the ITC binding studies and the Bi-Bi kinetic mechanism, the authors should consider that among GTs the basis for this kinetic mechanism is typically a steric/geometric one. The donor substrate is typically buried under the acceptor (and this should be examined in these structures) and productive catalysis can only occur when the donor substrate binds first not because it binds first. This is an example of where binding studies alone can be misleading. Enzyme kinetic analysis shows that high acceptor concentrations leads to substrate inhibition in these systems.

The last three sentences of the paragraph beginning with "Given the strict reaction order" are an attempt to explain some of the major findings of the structural and biochemical analysis but they are speculative and unclear at best. Is the binding affinity of Rubu, ST, STB and Reb E in the presence of UDP even relevant? Perhaps the β 2-glucose moiety is not energetically unfavorable in the presence of UDP-glucose during binding and catalysis – K_m and k_{cat} will shed light on this

question. What do the authors mean by non-productive binding to the enzyme-UDP complex by mistake (see below)?

The discussion section is particularly poorly written/organized. It begins with modelling studies and more binding data (4-nitrohenyl β -D-glucopyranoside), subject matter that would better have been dealt with in the results section. The paragraph beginning with "The order of R1 then R2 results from" is highly speculative and attempts to extrapolate from ground state structures and binding data to what must be happening in the transition state. As mentioned above, a measure of K_m and k_{cat} is required before attempts to rationalize the reaction rates and fine specificity can be made. The idea that certain acceptor structures might inhibit the enzyme by binding the UDP bound form of the enzyme is possible but predicated on the idea that the UDP concentration in the cell is high enough to maintain that complex. What basis is there for this supposition? The authors state that when Reb B, Reb A and Reb D bind with the R2 end in the active site no further binding energy can be gained (presumably from the sugars) and that this retards the reaction relative to that found at R1. However, as the authors point out the steviol binds in an entirely different orientation when R2 is in the active site. How do the authors rule out the possibility that differences in the enzyme-steviol interaction do not lead to the rate differences at R1 and R2?

In addition, the following should be noted:

- 1) The authors state that the ability to specifically generate β -1,3 linkages (they call this regioselective which in itself is confusing) at the different ends and/or different forms of the growing saccharides of steviol glycosides represents a paradox. However, virtually all glycosyltransferases generate a specific linkage and many can glycosylate the acceptor hydroxyl/residue/moiety in structurally different contexts. They also state that "This degree of plasticity in the mechanism of substrate recognition is novel in a regioselective enzyme". Do the authors suggest that there are GTs that use this degree of plasticity but that do not show linkage specificity? What about other degrees of plasticity? In any case, these statements are misleading if not wrong and seem to be aimed at somehow heightening the significance/impact of the work.
- 2) In a number of places the authors state that the enzyme shows an obligate or strict reaction order where the R1 end is glycosylated before the R2 end. At best they have shown that the R1 end is glycosylated at a higher rate than that of the R2 end.
- 3) The authors state that there is one molecule in the asymmetric unit because no stable multimer is identified by PISA and gel filtration shows a monomer. This need to be corrected.

Reviewer #3 (Remarks to the Author):

The manuscript is an interesting multidisciplinary work concerning the structure of UGT76G1 with different ligands together with HPCL/mass spectrometry and ITC experiments. The most interesting part of the article is that the diterpenoid steviol backbone is mainly the most important structural feature to achieve an optimal binding to the enzyme. In addition, this moiety should adopt two different binding modes in order for glycosylation to take place in both R1 and R2 positions. It is also interesting to mention that they disentangle what region of the acceptor substrate is firstly glycosylated.

Nevertheless, it is not surprising that this enzyme has some plasticity to bind the steviol backbone or the steviol glucoside because it is glycosylating two different positions of the compounds. The paper will improve considerably if they address the comments below and sort out the inconsistencies found along the text.

Major comments:

- Proper enzyme kinetics should be performed for at least STB, RebA and RebD. In addition, it

would be important to do kinetics for RebE and Rubu because it would be reasonable to get kinetic constants since the second reaction might proceed with a very slow turnover. In any case, they need to address the importance of the glycosylation of the second glycosylation site because it appears that the reaction would be extremely slow. If this turns to be true, this inherent plasticity of this enzyme is rather limited because the first glycosylation point is rather the important one in terms of catalysis and binding.

- A thorough discussion of the ITC experiments should be present in the manuscript. The authors do not discuss at all the K_d values. In addition, ITCs performed with UDPG are not entirely credible because UDPG will be hydrolysed during the time of the ITC experiment. This needs to be clarified in the revised version.

- Privateer should be used to check whether the sugars in the crystal structures have the right conformations.

- Nomenclature is not clear for the sugars in the text. It is difficult to follow the text according to the vague nomenclature used in the manuscript. They could use names for the sugars such as A, B and C and also AR1, etc, to indicate the position in the acceptor substrates.

- What is the paradox coming from? First, the enzyme appears to glycosylate just only two sites in the acceptor substrates. Therefore, it is not really very promiscuous. Then, the second glycosylation site appears to be a very poor glycosylation site and this could clearly suggest that it is not entirely a very promiscuous enzyme. In addition, it is clear that a different binding mode should take place to glycosylate the second glycosylation site with respect to the first glycosylation site. All these circumstances clearly diminish the claimed plasticity of this enzyme, limiting the novelties of this work.

- To really clarify the type of kinetic mechanism, they need to perform proper kinetics in order to demonstrate that the enzyme follows a Bi-Bi ordered mechanism.

Minor comments:

- For the second glycosylation reaction to take place, it is more reasonable to speculate that the incoming glucose at R1 would clearly have steric hindrance with the glucose of UDPG, and this will be the decisive force to rotate the diterpenoid moiety. This should be clarified in the revised version of the manuscript.

Answers to the Reviewers' comments

Reviewers' comments:

Reviewer #1 (Remarks to the Author):

The manuscript with the title "Hydrophobic recognition allows the glycosyltransferase UGT76G1 to catalyze the substrate in two orientations but retain regiospecificity" by Yang et al. describes the structural and functional characterization of an glycosyltransferase from *S. rebaudiana* involved in the production of the sweeteners RebM and RebD, which are preferred due to their specific characteristics in receptor activation.

The authors characterize the enzyme using activity assays, binding studies, and x-ray crystallography to unravel the unique characteristics of the enzyme in respect of substrate specificity and activity. Overall the manuscript is very interesting and the body of work is carried out well, however, at times it looks as the manuscript is put together a bit sloppy. It would benefit from an additional round of careful reading.

> we apologize and have restructured the manuscript to focus on the how flexibility in binding modes co-exists with site-specific catalysis.

Specific comments to the manuscript are outlined below.

Introduction:

Would be highly beneficial ☐ is highly beneficial

Glycosylic bonds ☐ glycosidic bonds

Microbial production of Reb M and Reb D present potential solutions

> We have corrected the language here.

It is mentioned that UGT76G1 is the only known enzyme to perform beta 1-3 glycosylation in *S. rebaudiana*. Are there other enzymes for the beta 1,2 addition?

> This is an interesting question that we are working on but is different from the subject of this paper. Apart from UGT91D2, there is no other enzyme that has been detected in *Stevia rebaudiana*. We cloned UGT91D2 but have not succeeded in overexpressing UGT91D2 in a soluble and active form

Results Though most GT'S are metal-dependent, there are well-documented examples, where this is not the case.and high-resolution mass spectrometry (MS) rather than spectroscopy and also the "analyses" can be removed

> The text has been modified as requested

Which other substrates besides the mentioned steviosides are glycosylated? One artificial one is mentioned, but the sentence implies there are more. Are these other steviosides ore something completely different? The mentioned one has been used in the paper.

> We have synthesized two new unnatural steviol glycoside substrates for the revision of the paper by using the enzymes UGT85C2 and UGT74G1 (see

the reaction scheme in Figure 1b). These new compounds have a single glucose which is attached at the R1 or R2 position respectively. We show that both are substrates of UGT76G1, establishing the following two key related points.

1 The R1 R2 order is a function of kinetics, there is no requirement for modification of the R1 site for R2 to become a substrate. Thus it should be stated as a preference (which we now do).

2 The ability to modify R2 is a function of the steviol moiety not due to the blockage at the R1 site.

We apologize for the lack of clarity in the original manuscript, there are other steviol glycosides and all have the same steviol aglycone ring (diterpenoid) but have different glycones. We have tested all the relevant steviol glycoside molecules that we can access and made two new (unnatural) molecules. A key finding was the demonstration that 4-nitrophenyl β -D-glucopyranoside is a substrate. The molecule is completely different from the steviol glycosides and it was used to test our prediction on the recognition rules.

Very interesting is the analysis of the order of the reaction. It seems quite obvious that the R1 site is preferred, but is there really a very strict order? There is unfortunately now stevioside with just the R2 site modified without sugars at the R1 site. Also, might it be that the RebE R1 mod and R2 mod run in HPLC at the same elution time? So a minor product of R2 modified first might be not really detectable over the majority of R1 modified first product.

> There is a very obvious preference for the R1 reaction over the R2 reaction from the new kinetic data (which meets in part what the reviewer requested).

In the original manuscript, we are unable to see any evidence for modification at R2 where R1 modification is possible based on HPLC. During the revision, Our data from the synthetic steviol monoglycoside containing a glucose at the R2 site and a bare (no sugar) R1 site, have established that the enzyme can indeed modify at the R2 site in the absence of sugar at R1.

We accept the referee's wider point, we don't imply the apparent order is 'obligate' that R1 MUST be processed first in some way. We have amended the text to talk about the obvious preference.

"There is only one molecule in the asymmetric unit since no stable multimer is identified using PISA. "

I'm afraid, but this statement is wrong. The oligomeric state and the content of the asymmetric unit are two different and unrelated things. There can be monomers in the au but the molecule is an oligomer in solution and in the crystal if symmetry related monomers are taken into account.

> This was a typo. We have corrected the manuscript to

“There is one molecule of UGT76G1 in the asymmetric unit and no stable symmetry related multimer is identified by the PISA web server²⁰. Gel filtration suggests UGT76G1 exists as a monomer in solution.”

Rmsd \approx r.m.s.d.

Central six-strand beta - \approx central six-stranded

> We have changed the text.

Generally speaking in the conformation observed, Reb A might be a product structure, but in principle it can undergo a second glycosylation at R2. The sentence should be rephrased to make that qualification.

> We have changed the text to

“The UDP Reb A structure can be thought of as representing the products formed by sugar transfer to the R1 position of ST”

One should be careful with a statement of the transition state modelling. There is still controversy about the exact mechanism and therefore concomitant transition state for quite some GT's despite extensive trials to clarify the mechanism.

>The reviewer is right that the transition state of the retaining glycosyltransferase is not certain.

The glycosyltransferase in this manuscript is an inverting enzyme where the dissociative SN₂ mechanism is we believe widely accepted.

We have however de-emphasized discussion of our models of the transition state.

From Figure 3 I cannot see or understand the claim that in Rubu the β 2 glucose would clash with His25 or Leu 204.

> We apologize. We meant the substrate containing the β (1-2) linked glucose could not adopt the same steviol aglycone binding orientation found in Rubu complex. This is because the β (1-2) linked glucose would clash with His25 or Leu 204. Thus the steviol aglycone has adjusted its orientation seen in the binding site between Reb A and Rubu.

In Figure 3, we showed the different binding orientation of aglycone ring of Reb A and Rubu that is found in crystal structures. We didn't show the clashes that would occur if the aglycone of the Reb A adopted the same orientation found in the Rubu complex.

That entire subparagraph is hard to understand and it is not clear what the authors want to say with that paragraph.

> We apologize and have removed this paragraph

The authors said they cannot get binding curves with UDPG due to reaction heat. That is correct but allows at the same time to get a K_m and k_{cat} value from the enzymatic reaction if the experiment is done correctly.

> We have now performed steady-state kinetic assays deriving k_{cat} and K_m reported. We have performed enzyme kinetics for the commercially available steviol glycoside substrates when UDPG is at the saturating concentration. For those substrates which undergo sugar transfers at both at R1 and R2 sites, we only measured the (preferred) single sugar transfer at R1 site by choosing a short reaction (consumption of the substrate is less than 10%). To characterize the R2 reaction we used substrates which could only be modified at this position.

Contributing entropic and enthalpic components to different states is tricky and might be not justified here Especially the interference of bonding statement is very confusing and it is not clear what the authors want to argue here.

> We have removed the ITC data when UDPG is present and also the discussion about the entropy or enthalpy from this revision. We agree it is not relevant to the main thrust of the paper.

For the bi-bi mechanism to work the authors say UDPG binds first and the steviol second. Therefore and in line with the conserved binding mode of UDP(G) and analogs between different families it seems unlikely that the binding of steviol will contribute significantly to changes in the binding of UDP by movements not related to residues involved in binding of the UDP(G)

> We agree with the reviewer and removed this statement that the binding of steviol feeds back to the binding of UDP.

It is noteworthy that despite an expected stoichiometric 1 to1 binding and a μM binding constant, the N value is often significantly below 1. That indicates problems in the model or the concentrations of the protein or ligand. Furthermore despite reasonably good binding constant the lower plateau is not at all visible in any of the binding isotherms

The paragraph arguing about binding of the substrates/ products and UDP and kinetically favoured bindings is again very hard to read and understand.

> The ITC data have been removed along with the statements and discussion based on ITC data. We think this has clarified the manuscript.

I think the product with multiple hydrogen bonds binds readily. Especially taking into account that there is also a rearrangement of loop regions of the protein. I what way the beta

2 glucose does contribute to the overall reaction is debateable and needs to be further clarified.

> The β (1-2) glucose accelerates the rates of both R1 and R2 reactions (kinetic data). The structure shows that the beta 2 glucose in the complex structure of Reb A makes a number of contacts with the protein which we suggest it optimizes the position of the reactive accepting glucose (glucose A in the revision).

Reb I shows no binding with UDPG this is presumably for steric reasons, but Reb I is a product and should anyway show very weak binding to an ! active complex. A titration of Reb I in UDP enzyme might be interesting If it is purely steric than Reb I should have quite good affinity to the enzyme

Reb B with UDP should give similar Kd values as ST STB and RebE. That would be a good and necessary control. ReB B doesn't contribute to a flipped binding mode. It simply binds that way due to steric clashes with UDPG according to the authors interpretation.

>The ITC data have been removed, as the reactivity profile of two additional synthesized substrates is solid to support the role of the aglycone ring in catalysis of both ends.

Discussion:

...include the hydrophobic stevil ring and a glucose

Overwhelm the importance? What is meant with that phrase?

> We have rephrased for clarity as

“Analysis of the crystal structures reveals that in the acceptor substrate, neither the sugar and nor the aglycone participates in a network of hydrogen bonds. Rather both moieties are held in place by hydrophobic (van der Waal) interactions.”

Reb B contribution: I wonder if the steviol binds on its own similar good as Reb B That would be a good experiment ot verify the claim and show unequivocally that this is indeed the case.

> The ITC has been removed and also the solubility of the bare steviol limits the measurements in good quality.

The whole paragraph discussing binding modes in the light of potential transition states and interactions explaining the reaction order and potential activation barrier is highly speculative and with the currently available data hardly justifiable. That should be rewritten and be founded more on the actual data.

> We apologize for the confusion and have deleted this paragraph and rewritten the discussion.

The idea presented to uses hydrophobic handles is a very interesting concept and it will be indeed very interesting to see if other enzymes does work in the same way.

> Yes, we agree this is an interesting future area of research.

The other glycosyltransferases, which can catalyze the sugar added to hydrophobic aglycone, may be those too can handle simple sugar-hydrophobic group type substrates, such as 4-nitrophenyl β -D-glucopyranoside.

Methods:

Data collection parameters indicate very good statistics. It is unclear why the data were cut off at the values chosen here. The CC1/2 I/sig I all indicate that the crystals diffracted to far higher resolution.

> We used DIALS to process the data and the cc(1/2) determined the resolution limit. The aimless log file for the Reb A complex structure is shown

```
#####
<!--SUMMARY_BEGIN $TEXT:Result: $$ $$
Summary data for          Project: AUTOMATIC Crystal: DEFAULT Dataset: NATIVE
Overall InnerShell OuterShell
Low resolution limit      42.88      42.88      1.74
High resolution limit     1.70      7.60      1.70
Rmerge (within I+/I-)    0.084     0.059     1.139
Rmerge (all I+ and I-)   0.084     0.059     1.150
Rmeas (within I+/I-)     0.088     0.062     1.201
Rmeas (all I+ & I-)     0.087     0.061     1.181
Rpim (within I+/I-)     0.027     0.019     0.380
Rpim (all I+ & I-)      0.020     0.014     0.266
Rmerge in top intensity bin 0.051     -         -
Total number of observations 1045877   11753    77476
Total number unique       54657     684      4025
Mean(I)/sd(I)             18.9      34.7     4.6
Mn(I) half-set correlation CC(1/2) 0.999     0.998     0.978
Completeness              100.0     98.8     100.0
Multiplicity               19.1      17.2     19.2
Anomalous completeness    100.0     98.7     100.0
Anomalous multiplicity     9.9       10.2     9.9
DelAnom correlation between half-sets -0.438    -0.477    -0.039
Mid-Slope of Anom Normal Probability 0.656     -         -
No significant anomalous signal

Estimates of resolution limits: overall
  from half-dataset correlation CC(1/2) > 0.30: limit = 1.70A == maximum
resolution
  from Mn(I/sd) > 1.50: limit = 1.70A == maximum
resolution

Estimates of resolution limits in reciprocal lattice directions:
  Along h k plane
    from half-dataset correlation CC(1/2) > 0.30: limit = 1.70A == maximum
resolution
    from Mn(I/sd) > 1.50: limit = 1.70A == maximum
resolution
  Along l axis
    from half-dataset correlation CC(1/2) > 0.30: limit = 1.70A == maximum
resolution
    from Mn(I/sd) > 1.50: limit = 1.77A
#####
```

Figures:

Figure 3:

I would suggest to go a bit more off white in Fig 3a to enhance the contrast for the backbone

> We agree and changed the color to a more off white for a better contrast.

Figure 3c and d would benefit from being stereo due to their complexity

> We agree and put the stereoviews in the supplemental Figure 12.

Figure 5a)

It is hard to give suggestions how to improve this figure. It is hard to see what belongs to what. One option would be to have the scheme (chemdraw) of the two steviol compounds next to it to help to navigate through that overlay.

> We have changed Figure 5 and placed a clearer version of Figure 5a in the supplemental information (Supplemental Figure 13).

Figure 5c)

Thought the scheme itself is nice, it suggests a considerable structural plasticity (domain movement), which does not take place. There is a rearrangement of a loop but not major movements. At least not described in the manuscript.

> We have changed the organization of Figure 5. The original Figure 5c was removed which seems speculative to suggest such domain movement based on the available data.

New cartoons were added to Figure 4 to show how flexibility in binding modes co-exists with site-specific catalysis.

Supplementary

Fig S1. This figure seems unnecessary. The activity assay is much clearer in regard of the reactivity of the enzyme in respect to metal ion content. Taken into account would be only important to coordinate the phosphate and neutralise the charge.

> We would like to keep this supplemental figure (now Supplemental Figure 2) since it supports a statement made (and referenced) in the manuscript. We accept it might be overkill so to speak.

Fig S2: Some of the labels are hardtop read as they overlap with the peaks. It might help to move them a bit.

> Agree and changed.

Fig. 9Sb could be also worth doing it in stereo.

> We have done.

Generally often analyses is used, instead of analysis. For example

Fig S12: It is MS and Ms/Ms analysis of the reactions.....

> Corrected

Figures S10 The H25A structure inlet overlays with the MS spectrum below. Furthermore I can't find an explanation of the inlet in the legend of the figure.

> We moved the inset figure to avoid the overlap and indicated the hydrogen bonds involved. We have added the explanation to the legend, and we apologize for the oversight.

The waters are discussed in the main text and it is important for the discussion. This part should go in one of the main figures to explain better how the waters compensate for the lack of the His side chain.

> There is trace activity for H25A. We did not want to distract from the manuscript by a long discussion. We have shortened the discussion to focus on the essential and highlighted the key water.

Reviewer #2 (Remarks to the Author):

This paper reports the x-ray structure of UGGT6G1 (and complexes) a glucosyltransferase involved in the biosynthesis of steviol glycosides, natural products that are used as sugar substitutes. It also reports experiments aimed at characterizing the molecular basis for the observed substrate binding and catalytic properties shown by this enzyme.

Steviol possesses a diterpenoid backbone which is glycosylated (up to trisaccharides) at both ends (ie. R1 and R2 end) to generate the various naturally occurring steviol glycosides. The main conclusion of the paper is that UGGT6G1 recognizes the steviol aglycone through apolar interactions that are not dependent on a unique structural mode of interaction. In this way, the enzyme can glucosylate the saccharide moiety at the R1 and R2 ends by binding to the steviol backbone in two different ways.

The x-ray structures have been determined at high resolution and based on the data collection and refinement statistics they seem to be of high quality. The ITC and rate measurements also seem to be sound.

Although the structures are novel and the system is an interesting one, the paper is lacking at a number of levels. Perhaps the greatest flaw is the lack of enzyme kinetic analysis. Any attempt to rationalize specificity differences such as the R1/R2 end preference and the role played by the β 1,2-linked glucose moiety require a measure of both K_m and k_{cat} for the various substrates.

> As suggested by the reviewer, we have added a steady-state kinetic analysis of the substrates ST, Rubu, Reb D, RebE and STB. We now present and discuss the K_m and k_{cat} data. These data do indeed address the preference for R1 vs R2 and the role of the β 1,2-linked glucose to enhance the catalysis.

We have performed enzyme kinetics for the commercially available steviol glycoside substrates when UDPG is at the saturating concentration. For those substrates which undergo sugar transfers at both at R1 and R2 sites, we only measured the (preferred) single sugar transfer at R1 site by choosing a short

reaction (consumption of the substrate is less than 10%). To characterize the R2 reaction we used substrates which could only be modified at this position.

In a number of places the material presented and/or discussed is highly speculative, poorly communicated and/or not central to the thesis of the paper. The discussion of the differences in enthalpy/entropy of binding shown by UDP and UDP-glucose provide an example. At the end of that section the sentence beginning with “The different energy profiles between” is incomprehensible.

> We have removed ITC data and the speculation about the enthalpy/entropy and the different energy profiles which are not central to the main conclusions.

In the revision, we have rewritten the discussion section and it is more focused on what controls the two alternative binding orientations for the reactions on both ends of the substrate.

With regard to the ITC binding studies and the Bi-Bi kinetic mechanism, the authors should consider that among GTs the basis for this kinetic mechanism is typically a steric/geometric one. The donor substrate is typically buried under the acceptor (and this should be examined in these structures) and productive catalysis can only occur when the donor substrate binds first not because it binds first. This is an example of where binding studies alone can be misleading. Enzyme kinetic analysis shows that high acceptor concentrations leads to substrate inhibition in these systems.

>We have removed the ITC data.

We tried to use the classic product inhibition (not dead-end analog inhibition) to derive the kinetic mechanism (bi-bi etc). The assays can only be done discontinuously (and are technically challenging) and although we tried to accurately quantify product yield, the data are not of good enough quality to be unambiguous. Cautioned that the mechanism derived from the steady-state assays is sometimes biased by the data quality or the mechanism preferred by the authors. [See the refs]

Structure **25**, 1034-1044. (2017) (the first paragraph on P1037)

Archives of Biochemistry and Biophysics **564**, 120–127. (2014) (the first paragraph of the right column on p125).

We have removed discussion of the ordered kinetic mechanism as a result.

In our study, we didn't observe any obvious substrate inhibition.

The last three sentences of the paragraph beginning with “Given the strict reaction order” are an attempt to explain some of the major findings of the structural and biochemical analysis but they are speculative and unclear at best.

> We have rewritten this section and discussed the apparent reaction order based on the kinetic data which show a preference for R1 over R2 (where both are available). We change “the strict reaction order” to “preference”.

We have synthesized two new unnatural steviol glycoside substrates by using the enzymes UGT85C2 and UGT74G1 (see the reaction scheme in Figure 1b). These new compounds have a single glucose which is attached at the R1 or R2 position respectively. We show that both are substrates of UGT76G1, establishing that the sugar transfer at the R1 position is not an essential prerequisite for the transfer to the R2 position. The R1 R2 order is a function of kinetics, there is no requirement for modification of the R1 site for R2 to become a substrate. Thus it should be stated as a preference (which we now do).

Is the binding affinity of Rubu, ST, STB and Reb E in the presence of UDP even relevant?

> This section based on ITC data has been removed.

Perhaps the β -2-glucose moiety is not energetically unfavorable in the presence of UDP-glucose during binding and catalysis – K_m and k_{cat} will shed light on this question.

> The K_m and k_{cat} of these substrates has been determined and shows that the β -2-glucose accelerates catalysis at both R1 and R2 ends.

What do the authors mean by non-productive binding to the enzyme-UDP complex by mistake (see below)?

>The preferred binding mode for those steviol compounds is with R1 at the active site. Some compounds such as Reb A cannot bind with R1 end at the active site when UDPG is bound (the glucose of Reb A and UDPG would overlap and clash). In this case, Reb A can only bind in its flipped form, with R2 at the active site and undergo catalyzed glucose transfer. However, with only UDP bound, Reb A has no clash and can thus stay in the preferred R1 orientation, which is not productive for R2 reaction.

The discussion section is particularly poorly written/organized. It begins with modelling studies and more binding data (4-nitrophenyl β -D-glucopyranoside), subject matter that would better have been dealt with in the results section.

> We have rewritten the discussion based on the new kinetic data, the reactions of the synthetic substrates, structural and biochemical data. We have re-organized the material. The discussion is more focused on what controls the two alternative binding orientations for the reactions on both ends of the substrate.

The results of 4-nitrophenyl β -D-glucopyranoside are now in the results section.

The paragraph beginning with “The order of R1 then R2 results from” is highly speculative and attempts to extrapolate from ground state structures and binding data to what must be happening in the transition state. As mentioned above, a measure of K_m and k_{cat} is required before attempts to rationalize the reaction rates and fine specificity can be made.

> We have removed the speculative explanation about the “The order of R1 then R2 results from”.

We rewrote this part about the reaction order in the point of view of the K_m and k_{cat} . The massive difference of k_{cat}/K_m between R1 and R2 reactions results in the apparent order of R1 then R2. However, based on the reactivity of the two new synthesized substrates, R1 reaction is not an essential condition for R2 reaction.

The idea that certain acceptor structures might inhibit the enzyme by binding the UDP bound form of the enzyme is possible but predicated on the idea that the UDP concentration in the cell is high enough to maintain that complex. What basis is there for this supposition?

> We have no evidence, so have removed this speculation

The authors state that when Reb B, Reb A and Reb D bind with the R2 end in the active site no further binding energy can be gained (presumably from the sugars) and that this retards the reaction relative to that found at R1. However, as the authors point out the steviol binds in an entirely different orientation when R2 is in the active site. How do the authors rule out the possibility that differences in the enzyme-steviol interaction do not lead to the rate differences at R1 and R2?

> The kinetic data show the K_m is higher for all R2 substrates and the k_{cat} lower. We would agree with the reviewer that the altered binding is most likely the reason for this.

“We presume that the altered interactions in this flipped form are responsible for the higher K_m and lower k_{cat} values seen for substrates processed at the R2 end.”

In addition, the following should be noted:

1) The authors state that the ability to specifically generate β -1,3 linkages (they call this regioselective which in itself is confusing) at the different ends and/or different forms of the growing saccharides of steviol glycosides represents a paradox.

>We have clarified what we mean and in the revision.

The regioselectivity here means the enzyme specifically recognizes and performs the sugar transfer to the 3-hydroxyl group only of glucose.

“The enzyme consistently creates a β (1-3) linkage as a paradox by pinpointing the 3-hydroxyl of the glucoses AR1 and AR2 but neglects their different structural contexts, in which the glucoses AR1 and AR2 are at the asymmetric ends of the diterpenoid steviol ring.”

The enzyme requires the aglycone portion to be attached to the glucose, simple sugar is NOT a substrate. The aglycone serves as a handle; so far so normal perhaps one might say. However, the enzyme is able to modify a glucose attached to the other end of the aglycone. This means that the enzyme is able to process a chemically different substrate (the aglycone is not symmetrical). We are not aware of any other example of this behavior of substrate flipping.

What gives UGT76G1 this combination of specificity and flexibility (paradox) is an interesting question scientifically and commercially (critical to the favorite sweet flavor).

However, virtually all glycosyltransferases generate a specific linkage and many can glycosylate the acceptor hydroxyl/residue/moiety in structurally different contexts. They also state that “This degree of plasticity in the mechanism of substrate recognition is novel in a regioselective enzyme”. Do the authors suggest that there are GTs that use this degree of plasticity but that do not show linkage specificity? What about other degrees of plasticity? In any case, these statements are misleading if not wrong and seem to be aimed at somehow heightening the significance/impact of the work.

>Not our intention and we apologize. The enzyme processes a substrate that depends on having a sugar attached to a “handle”. Such sugar handle type substrates are common of course. We have not identified any other example where the “handle” can be flipped and still bind thus support catalysis. This makes UGT76G1 unique.

2) In a number of places the authors state that the enzyme shows an obligate or strict reaction order where the R1 end is glycosylated before the R2 end. At best they have shown that the R1 end is glycosylated at a higher rate than that of the R2 end.

> We agree and we use the term “preferred” now.

3) The authors state that there is one molecule in the asymmetric unit because no stable multimer is identified by PISA and gel filtration shows a monomer. This need to be corrected.

> Apologies, typo corrected

“There is one molecule of UGT76G1 in the asymmetric unit and no stable symmetry related multimer is identified by the PISA web server²⁰. Gel filtration suggests UGT76G1 exists as a monomer in solution.”

Reviewer #3 (Remarks to the Author):

The manuscript is an interesting multidisciplinary work concerning the structure of UGT76G1 with different ligands together with HPCL/mass spectrometry and ITC experiments. The most interesting part of the article is that the diterpenoid steviol backbone is mainly the most important structural feature to achieve an optimal binding to the enzyme. In addition, this moiety should adopt two different binding modes in order for

glycosylation to take place in both R1 and R2 positions. It is also interesting to mention that they disentangle what region of the acceptor substrate is firstly glycosylated.

Nevertheless, it is not surprising that this enzyme has some plasticity to bind the steviol backbone or the steviol glucoside because it is glycosylating two different positions of the compounds.

The paper will improve considerably if they address the comments below and sort out the inconsistencies found along the text.

Major comments:

- Proper enzyme kinetics should be performed for at least STB, RebA and RebD. In addition, it would be important to do kinetics for RebE and Rubu because it would be reasonable to get kinetic constants since the second reaction might proceed with a very slow turnover.

>

We have performed enzyme kinetics for the commercially available steviol glycoside substrates when UDPG is at the saturating concentration. For those substrates which undergo sugar transfers at both at R1 and R2 sites, we only measured the (preferred) single sugar transfer at R1 site by choosing a short reaction (consumption of the substrate is less than 10%). To characterize the R2 reaction we used substrates which could only be modified at this position.

In any case, they need to address the importance of the glycosylation of the second glycosylation site because it appears that the reaction would be extremely slow. If this turns to be true, this inherent plasticity of this enzyme is rather limited because the first glycosylation point is rather the important one in terms of catalysis and binding.

> R1 reaction is preferred than R2 reaction shown by k_{cat}/K_m comparison.

The R2 reaction is NOT marginal, the kinetics parameters for Reb D are within a factor of 10 of that for the R1 reaction of STB (and Rubu).

The significance of the finding is that it is the transfer of the sugar to the R2 site which is critical to the favorite sweet flavor.

- A thorough discussion of the ITC experiments should be present in the manuscript. The authors do not discuss at all the K_d values. In addition, ITCs performed with UDPG are not entirely credible because UDPG will be hydrolysed during the time of the ITC experiment. This needs to be clarified in the revised version.

> Prompted by the reviewer's well-articulated concerns, we changed our approach removing the ITC and performing kinetics. The discussion has been rewritten based on the new kinetic data, the reactions of the synthetic substrates, structural and biochemical data. We have re-organized the material. The discussion is more focused on what controls the two alternative binding orientations for the reactions on both ends of the substrate.

- Privateer should be used to check whether the sugars in the crystal structures have the right conformations.

> we used privateer to check the quality of the sugar model and found both Reb A and Rubu are in right conformation. This analysis is reported in the revision.

Name	Chain	Q ¹	Phi	Theta	Anomer	D/L ²	Conformation	RSCC ³	<Bfactor>	Diagnostic
AQ9	A	0.553	213.549	4.13313	beta	D	⁴ C ₁	0.89	45.3567	Ok

¹Q is the total puckering amplitude, measured in Angstroms.

²Whenever N is displayed in the D/L column, it means that Privateer has been unable to determine the handedness based solely on the structure.

³RSCC, short for Real Space Correlation Coefficient, measures the agreement between model and positive omit density. An RSCC below 0.8 is typically considered poor.

- Nomenclature is not clear for the sugars in the text. It is difficult to follow the text according to the vague nomenclature used in the manuscript. They could use names for the sugars such as A, B and C and also AR1, etc, to indicate the position in the acceptor substrates.

> This advice is very useful and we used this suggestion.

- What is the paradox coming from? First, the enzyme appears to glycosylate just only two sites in the acceptor substrates. Therefore, it is not really very promiscuous. Then, the second glycosylation site appears to be a very poor glycosylation site and this could clearly suggest that it is not entirely a very promiscuous enzyme.

In addition, it is clear that a different binding mode should take place to glycosylate the second glycosylation site with respect to the first glycosylation site. All these circumstances clearly diminish the claimed plasticity of this enzyme, limiting the novelties of this work.

> The regioselectivity here means the enzyme specifically recognizes and performs the sugar transfer to the 3-hydroxyl group only of glucose.

“The enzyme consistently creates a β (1-3) linkage as a paradox by pinpointing the 3-hydroxyl of the glucoses AR1 and AR2 but neglects their different structural contexts, in which the glucoses AR1 and AR2 are at the asymmetric ends of the diterpenoid steviol ring.”

The enzyme requires the aglycone portion to be attached to the glucose, simple sugar is NOT a substrate. The aglycone serves as a handle; so far so normal perhaps one might say. However, the enzyme is able to modify a glucose attached to the other end of the aglycone. This means that the enzyme is able to process a chemically different substrate (the aglycone is not symmetrical). We are not aware of any other example of this behavior of substrate flipping.

What gives UGT76G1 this combination of specificity and flexibility (paradox) is an interesting question scientifically and commercially (critical to the favorite sweet flavor).

Our paper addresses this question and shows how it might be exploited with the nitrophenol handle.

- To really clarify the type of kinetic mechanism, they need to perform proper kinetics in order to demonstrate that the enzyme follows a Bi-Bi ordered mechanism.

> We tried to use the classic product inhibition (not dead-end analog inhibition) to derive the kinetic mechanism (bi-bi etc). The assays can only be done discontinuously (and are technically challenging) and although we tried to accurately quantify product yield, the data are not of good enough quality to be unambiguous. Cautioned that the mechanism derived from the steady-state assays is sometimes biased by the data quality or the mechanism preferred by the authors. [See the refs]

Structure **25**, 1034-1044. (2017) (the first paragraph on P1037)

Archives of Biochemistry and Biophysics **564**, 120–127. (2014) (the first paragraph of the right column on p125).

We have removed discussion of the ordered kinetic mechanism as a result.

Minor comments:

- For the second glycosylation reaction to take place, it is more reasonable to speculate that the incoming glucose at R1 would clearly have steric hindrance with the glucose of UDPG, and this will be the decisive force to rotate the diterpenoid moiety. This should be clarified in the revised version of the manuscript.

> We have synthesized two new unnatural steviol glycoside substrates for the revision of the paper by using the enzymes UGT85C2 and UGT74G1 (see the reaction scheme in Figure 1b). These new compounds have a single glucose which is attached at the R1 or R2 position respectively.

We show that both are substrates of UGT76G1, establishing the following two key related points.

1 The R1 R2 order is a function of kinetics, there is no requirement for modification of the R1 site for R2 to become a substrate. Thus it should be stated as a preference (which we now do).

2 The ability to modify R2 is a function of the steviol moiety not due to the blockage at the R1 site.

REVIEWERS' COMMENTS:

Reviewer #1 (Remarks to the Author):

Review NatComm manuscript NCOMMS-18-34404A „Hydrophobic recognition allows the glycosyltransferase UGT76G1 to catalyze the substrate in two orientations

The manuscript is a revision of a manuscript submitted earlier. The manuscript deals with the structural and functional characterisation of a glycosyltransferase from *Stevia rebaudiana* involved in the regiospecific glycosylation of steviol derivatives.

For the revision, the authors made substantial additional experiments and rewritings, which helped to clarify a number of issues raised.

Overall the substantial changes have improved the manuscript considerably and only a few small issues have to be addressed before publication.

Results:

Overall Structure:

,which consists two Rossmann-like fold domains at the N- and C-termini. ◊ The N- and C-terminal part could be removed. It is clear from the fact there are two domains, that one is N- and one is C-terminal.

Recognition of AR1:

It is written that AR1 in ST is recognized by F22 and L379. It seems from Fig 3 that the distance between AR1 and F22 is quite large and the angle is not quite right for Pi-stacking. L379 seems not to play any role in RubU recognition, rather I90 might be more important? What is

In figure 3d L85 is mislabelled as I85.

Based on which substructure is the R2 binding mode modelled? It seems that the binding site open up a bit with larger substrates. Is that maybe also needed for the flipped binding mode?

Discussion:

It might be worth to point out in that the results presented here are based on in vitro experiments. There are only trace amounts of RebD and RebM produced in the plant. Does that mean compared to the other enzymes this GT is rather low in activity? Maybe as a result of the rather high plasticity?

Based on Fig 1., the synthesis of RebM is from RebE. The most abundant steviosides are ST and RebA. RebA is probably produced from ST, but how does that fit in the scheme outline in Fig 1b.? RebD is certainly a good substrate, but still a factor of 10 different to the most likely "true" substrate. Depending on the location of the enzyme in the compartments and the concentrations of the respective steviosides, how likely is that the enzymes has evolved to be promiscuous? Maybe it has not lost the promiscuity during evolution, due to the lack of pressure thanks to compartmentalisation and substrate availability.

Nevertheless, that doesn't diminish the value of this promiscuity, especially for potential applications. The findings in the paper are of considerable interest.

Are held in place by hydrophobic (van der Waal) ◊ van der Waals

Normally seen for acceptor substrate recognition in glycosyltransferases. ◊ one or two examples could be cited to emphasize the point.

Both hydrogen bonds and van der Waal interactions, ◊ van der Waals interaction

Methods

Pilatus CCD: Pilatus is rather a pixel array and not a CCD type detector

Figures:

In figure 3d L85 is mislabelled as I85. The same error is than propagated to supplemental Figure 12b.

In Sup Fig 12 a label F22 is misplaced and only in one half of the Figure at all

Supplemental Figure 12 is somehow strange to look at. Is that wall eye or by accident cross eye? It seems in Fig a12a that I203 is colliding with the Stevioside?

Reviewer #2 (Remarks to the Author):

The authors have done a good job at addressing the reviewer's concerns and the paper is much improved both in terms of the strength of the science and the clarity of the presentation.

The following points still need to be addressed:

1) The kinetics work is a critical component of this paper and the raw data from the UDP-Glo assay needs to appear in the supplemental data section. The fits to the data used in calculating the kinetic parameters also needs to be shown.

2) The enzyme consistently creates a β (1-3) linkage as a paradox by pinpointing the 3-hydroxyl of the glucoses AR1 and AR2 but neglecting their different structural contexts, in which the glucoses AR1 and AR2 are at the asymmetric ends of the diterpenoid steviol ring.

By combining multiple structures of UGT76G1 with the reactivity profile of the available authentic and two self-made steviol glucosides, we have revealed the molecular basis of the paradox.

The use of the word "paradox" in these statements remains a source of confusion and should be removed. Indeed, removing the words "consistently", "paradox" and "pinpointing" will greatly improve the chemistry/enzymology.

3) The authors use the words "aglycone", "handle" and "tag" almost interchangeably. This must be addressed. Ideally, aglycone alone will be used.

4) Most glycosyltransferases require Mg^{2+} or other divalent metal ions²⁰, This is not true. Essentially all GT-B glycosyltransferases are metal ion-independent.

5) UGT76G1 is a type B-folded glycosyltransferase²⁰

The authors should use standard nomenclature: "GT-B fold glycosyltransferase"

6) followed by a dissociative SN_2 reaction

The word "dissociative" does not describe an SN_2 reaction and must be removed

Reviewer #3 (Remarks to the Author):

The revised version of the manuscript has been significantly improved and the authors have successfully addressed all my comments. The paper has been almost completely rewritten and they have performed proper enzyme kinetics addressing one of my major concerns. Overall, the current revised version of the manuscript is suitable for publication in NCOMMS.

Answers to the Reviewers' comments

Reviewers' comments:

Reviewer #1 (Remarks to the Author):

Review NatComm manuscript NCOMMS-18-34404A „Hydrophobic recognition allows the glycosyltransferase UGT76G1 to catalyze the substrate in two orientations

The manuscript is a revision of a manuscript submitted earlier. The manuscript deals with the structural and functional characterisation of a glycosyltransferase from *Stevia rebaudiana* involved in the regiospecific glycosylation of steviol derivatives.

For the revision, the authors made substantial additional experiments and rewritings, which helped to clarify a number of issues raised.

Overall the substantial changes have improved the manuscript considerably and only a few small issues have to be addressed before publication.

Results:

Overall Structure:

,which consists two Rossmann-like fold domains at the N- and C-termini. \diamond The N-and C-termin part could be removed. It is clear from the fact there are two domains, that one is N- and one is C-terminal.

> We prefer to keep “the N- and C-termini” to help describe the difference of two domains clearly.

Recognition of AR1:

It is written that AR1 in ST is recognized by F22 and L379. It seems from Fig 3 that the distance between AR1 and F22 is quite large and the angle is not quite right for Pi-stacking. L379 seems not to play any role in RubU recognition, rather I90 might be more important?

> We didn't mean to imply that AR1 is recognized by F22 and L379. Our point was that AR1 is sandwiched by these two residues. We changed the pi-stacking of F22 to the hydrophobic patch and included the contribution of Ile 90. The closest distance between the carbons of AR1 of Reb A and L379, F22, and I90 is 4.4 Å, 3.8 Å and 4.4 Å respectively, while the equivalent distance between AR1 of Rubu and L379, F22, and I90 is 4.5 Å, 3.8 Å and 4.4 Å respectively.

“Interestingly, glucose AR1 lacks extensive hydrogen bond recognition and is surrounded by Leu 379, Phe 22 and Ile 90 (hydrophobic patches) with only a single hydrogen bond between the 3-hydroxyl oxygen and the side chain of His 25.

What is In figure 3d L85 is mislabelled as I85.

>We apologize and corrected the mislabeling.

Based on which substructure is the R2 binding mode modelled? It seems that the binding site open up a bit with larger substrates. Is that maybe also needed for the flipped binding mode?

> We used the nonreactive Reb A substructure that has an ordered R2 glycone to model the R2 binding mode. The 3-hydroxyl of the glucose is positioned appropriately for catalysis by His 25. The "flipped" steviol aglycone

occupies a similar position of the aglycone ring of the Reb A and Rubu in the R1 binding mode structures.

We agree with the reviewer that the substrate binding is not like a snug-fit binding and the site may accommodate larger substrates than steviol substrates, but we are not sure if a larger site (or how large site) is required for the flipped binding mode.

Discussion:

It might be worth to point out in that the results presented here are based on in vitro experiments.

> **We agree with the reviewer and the readers would understand we are reporting the *in vitro* studies. We think it is fine to keep the manuscript as is.**

There are only trace amounts of RebD and RebM produced in the plant. Does that mean compared to the other enzymes this GT is rather low in activity? Maybe as a result of the rather high plasticity?

> **This is an interesting question for the future study of synthetic biology. It would be our guess that Reb D and Reb M are limited by the relatively slower R2 reaction and/or the slow rate of the β (1-2) glycosidic bond formation catalyzed by UGT91D2.**

Based on Fig 1., the synthesis of RebM is from RebE. The most abundant steviosides are ST and RebA. RebA is probably produced from ST, but how does that fit in the scheme outline in Fig 1b.?

> **The aim of figure 1b is to show the reaction catalyzed by each UGT in the biosynthesis at both R1 and R2 sites, we didn't intend it to show the abundance of the steviol glycosides.**

RebD is certainly a good substrate, but still a factor of 10 different to the most likely "true" substrate. Depending on the location of the enzyme in the compartments and the concentrations of the respective steviosides, how likely is that the enzymes has evolved to be promiscuous? Maybe it has not lost the promiscuity during evolution, due to the lack of pressure thanks to compartmentalisation and substrate availability.

> **Yes, this is a very good question and we do not have any experimental data to answer this. The evolutionary journey to this point will be of considerable interest for future study.**

"In the case of UGT76G1, nature has created an enzyme that can process different substrates by relying on hydrophobic recognition, which allows the synthesis of different glycans with the same shared set of enzymes."

Nevertheless, that doesn't diminish the value of this promiscuity, especially for potential applications. The findings in the paper are of considerable interest.

Are held in place by hydrophobic (van der Waal) \diamond van der Waals

> **Corrected**

Normally seen for acceptor substrate recognition in glycosyltransferases. \diamond one or two examples could be cited to emphasize the point.

> **Reference added**

Both hydrogen bonds and van der Waal interactions, \diamond van der Waals interaction

> Corrected

Methods

Pilatus CCD: Pilatus is rather a pixel array and not a CCD type detector

> Corrected

Figures:

In figure 3d L85 is mislabelled as I85. The same error is than propagated to supplemental Figure 12b.

In Sup Fig 12 a label F22 is misplaced and only in one half of the Figure at all

> Corrected

Supplemental Figure 12 is somehow strange to look at. Is that wall eye or by accident cross eye? It seems in Fig a12a that I203 is colliding with the Stevioside?

> Checked and make sure the figures in the wall-eye stereo.

Reviewer #2 (Remarks to the Author):

The authors have done a good job at addressing the reviewer's concerns and the paper is much improved both in terms of the strength of the science and the clarity of the presentation.

The following points still need to be addressed:

1) The kinetics work is a critical component of this paper and the raw data from the UDP-Glo assay needs to appear in the supplemental data section. The fits to the data used in calculating the kinetic parameters also needs to be shown.

> All original data and the fits are included in the source data.

2) The enzyme consistently creates a β (1-3) linkage as a paradox by pinpointing the 3-hydroxyl of the glucoses AR1 and AR2 but neglecting their different structural contexts, in which the glucoses AR1 and AR2 are at the asymmetric ends of the diterpenoid steviol ring.

By combining multiple structures of UGT76G1 with the reactivity profile of the available authentic and two self-made steviol glucosides, we have revealed the molecular basis of the paradox.

The use of the word "paradox" in these statements remains a source of confusion and should be removed. Indeed, removing the words "consistently", "paradox" and "pinpointing" will greatly improve the chemistry/enzymology.

> We have removed these words.

3) The authors use the words "aglycone", "handle" and "tag" almost interchangeably. This must be addressed. Ideally, aglycone alone will be used.

> We have mainly used "aglycone", we have removed the word "tag". We have kept the word "handle" only when conveying the concept and linking to other systems (sugar nucleotide).

4) Most glycosyltransferases require Mg^{2+} or other divalent metal ions²⁰, This is not true. Essentially all GT-B glycosyltransferases are metal ion-independent.

> Corrected

5) UGT76G1 is a type B-folded glycosyltransferase²⁰

The authors should use standard nomenclature: "GT-B fold glycosyltransferase"

> **Corrected**

6) followed by a dissociative SN2 reaction

The word "dissociative" does not describe an SN2 reaction and must be removed

> **Corrected**

Reviewer #3 (Remarks to the Author):

The revised version of the manuscript has been significantly improved and the authors have successfully addressed all my comments. The paper has been almost completely rewritten and they have performed proper enzyme kinetics addressing one of my major concerns. Overall, the current revised version of the manuscript is suitable for publication in NCOMMS.